# ReFocusEraser: Refocusing for Small Object Removal with Robust Context-Shadow Repair

**Qingping Zheng**[1,2,*]     **Bo Huang**[3,*]     **Yang Liu**[4]     **Haoyu Zhao**[5]     **Ling Zheng**[6]
**Zengmao Wang**[7]     **Ying Li**[3]     **Jiankang Deng**[8]

[1]School of Informatics, Xiamen University
[2]Key Laboratory of Multimedia Trusted Perception and Efficient Computing,
Ministry of Education of China, Xiamen University
[3]School of Computer Science, Northwestern Polytechnical University
[4]Department of Engineering, King's College London
[5]Institute of Trustworthy Embodied AI, Fudan University
[6]Fujian Maternal and Child Health Hospital, Fujian, China
[7]School of Computer Science, Wuhan University    [8]MVP Lab, Imperial College London
zhengqingping@xmu.edu.cn   bohuang@mail.nwpu.edu.cn
yang.15.liu@kcl.ac.uk   hyzhao22@m.fudan.edu.cn
zhengling@fjsfy.com   wangzengmao@whu.edu.cn
lybyp@nwpu.edu.cn   j.deng16@imperial.ac.uk

## Abstract

Existing diffusion-based object removal and inpainting methods often fail to re-cover the fine structural and textural details of small objects. This is primarily due to the VAE encoder's downsampling, which inevitably compresses small masked regions and causes significant detail loss, while the decoder's upsampling alone cannot fully restore the lost fine details. However, the adverse effects of this fixed compression can be mitigated by enlarging the perspective of these regions. To this end, we propose **ReFocusEraser**, a two-stage framework for small object re-moval that combines camera-adaptive zoom-in inpainting with robust context- and shadow-aware repair. In Stage I, a camera-adaptive refocus mechanism magni-fies masked regions, and a LoRA-tuned diffusion model ensures precise semantic alignment for accurate reconstruction. However, reintegrating these magnified in-painted regions into the original image introduces challenges due to VAE asymme-try, such as color shifts and seams. Stage II addresses these issues by fine-tuning an additional decoder to create a seam- and shadow-aware module that elimi-nates residual artifacts while preserving background consistency. Extensive ex-periments demonstrate that our proposed ReFocusEraser achieves state-of-the-art performance, outperforming existing methods across benchmark datasets. Related code and data are available at https://github.com/ProAirVerse/ReFocusEraser.git.

## 1    Introduction

Image object removal seeks to fill user-specified regions with content that is both visually and seman-tically consistent Bertalmio et al. (2000). While diffusion-based methods Xie et al. (2023) achieve state-of-the-art results, producing realistic and coherent restorations, they often favor generating new objects over completing backgrounds. To address this, recent approaches introduce explicit guidance to better align generated content with intended semantics and maintain visual coherence, including sampling known pixels during reverse diffusion Lugmayr et al. (2022), leveraging Alpha-CLIP Sun et al. (2024) embeddings to prioritize background regions Ekin et al. (2024), employing task and negative prompts for controllable inpainting Zhuang et al. (2024), separating foreground and background embeddings Xu et al. (2025), or incorporating both foreground and background images as independent guidance signals Wei et al. (2025).

---

*Equal Contribution. Correspondence to Qingping Zheng (zhengqingping@xmu.edu.cn)

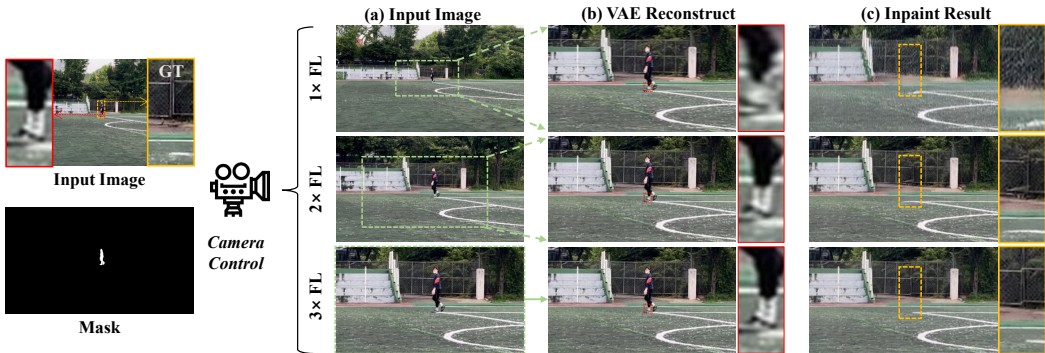

Figure 1: **Illustration of camera-adaptive refocus.** Far-field regions are zoomed in near-field views, enlarging small masked objects and preserving details during inpainting. 'FL' is focal length.

However, existing diffusion-based object removal methods struggle to accurately restore fine structural and textural details of small unwanted objects. This limitation primarily arises from the downsampling layers in the VAE encoder of these generative models Peebles & Xie (2023a); Esser et al. (2024), which inherently cause information loss. Specifically, under a fixed VAE compression rate, large masked objects can retain most of their structure, whereas small masked objects suffer from severe detail loss Rombach et al. (2022). This effect is evident in Figure 1(b), where high-frequency details of small objects are largely discarded during encoding, leading to noticeably degraded reconstruction quality. Although finetuning the entire VAE can mitigate this issue, it comes at a high computational cost, requires large-scale training data, and is prone to unstable optimization that may incur posterior collapse Lucas et al. (2019), ultimately reducing the diversity of generated outputs.

Intuitively, increasing the input resolution can alleviate this problem Wu et al. (2025). Motivated by camera zoom mechanisms in real-world photography, far-field views can be transformed into near-field perspectives through focal length adjustment, effectively bringing distant subjects closer and making them appear larger. Following this principle, we can simulate zoomed-in views to refocus the image, enlarging small masked objects and enhancing detail preservation prior to inpainting. As demonstrated in Figure 1, this camera-adaptive refocus strategy yields substantial improvements in reconstruction quality for small regions. Since this process improves inpainting quality only within the magnified region, the result must be zoomed out and pasted back into the original image. However, this step is challenging: the inherent asymmetry between VAE downsampling and upsampling often introduces color shifts Wang et al. (2025), leading to visible seam artifacts.

In this paper, we propose ***ReFocusEraser*** a framework for small masked object removal with robust context and shadow repair. Given an image with a user-defined mask, we first transform far-field regions into near-field perspectives using a camera calibration algorithm Veicht et al. (2024) to avoid distortion. Then, we introduce LoRA to fine-tune the FLUX.1-dev's diffusion transformer Black-Forest-Labs (2024) by minimizing the reconstruction loss of the enlarged masked object, improving semantic alignment between the inpainted region and its surrounding context. To maintain background consistency completely, we adopt mask-based stitching to paste the zoomed-out inpainted region back into the original masked image. However, this approach can introduce color discrepancies and fails to remove shadows of small objects that are not explicitly masked. To address these issues, we keep the FLUX.1-dev encoder unchanged and fine-tune an additional decoder to create a seam- and shadow-aware decoder. This enables seamless integration of the inpainted region and automatic shadow correction before reintegration into the original masked image. In summary, our main contributions are as follows:

- We introduce a *Camera-Adaptive Refocus Inpainter* which enlarges small masked objects via simulated near-field perspectives and improves semantic consistency by fine-tuning the diffusion model with LoRA on the enlarged regions.

- We develop a *Seam- and Shadow-Aware Repair* that seamlessly integrates inpainted regions and automatically corrects shadows of small objects while using a mask-based stitching strategy to robustly preserve background context.

- We propose ReFocusEraser, a two-stage framework for both small and large object removal, surpassing prior state-of-the-art methods (PSNR/SSIM/FID: 31.26/0.924/21.38 vs. 24.06/0.656/25.39 for AttentiveEraser and 25.03/0.794/65.22 for Flux-Fill).

## 2 RELATED WORK

**Image Inpainting.** Image inpainting aims to restore missing or corrupted regions with visually and semantically coherent content. Traditional methods Criminisi et al. (2004); Zhang et al. (2018a); Barnes et al. (2009) based on texture synthesis or patch matching often fail in complex scenes due to limited structural and global reasoning. GAN-based approaches Zhao et al. (2021); Li et al. (2022); Suvorov et al. (2022) improve realism via learned priors but suffer from artifacts and unstable training. Diffusion-based inpainting has recently emerged as a powerful alternative due to its strong generative capacity and controllability. Early works Avrahami et al. (2023); Yang et al. (2023); Zhang et al. (2023); Lugmayr et al. (2022); Liu et al. (2023); Corneanu et al. (2024) sample masked regions using pre-trained text-to-image models, achieving impressive results but often lacking semantic alignment in complex scenes. To improve structure and content awareness, later methods fine-tune diffusion models with explicit mask conditioning in latent space.

**Mask-Conditioned Diffusion Inpainting.** To mitigate this limitation, diffusion-based inpainting methods typically require explicit guidance (e.g. prompts, structural cues, or contextual priors) to maintain semantic accuracy and visual consistency. For instance, RePaint Lugmayr et al. (2022) guides generation with masked input, sampling known pixels during reverse diffusion to improve semantic consistency using the DDPM Song et al. (2020) prior. CLIPAway Ekin et al. (2024) uses AlphaCLIP Sun et al. (2024) embeddings to separate foreground and background, prioritizing the background to enhance inpainting quality and reduce object hallucination. PowerPaint Zhuang et al. (2024) uses task-specific prompts, including negative prompts and controllable shape-fitting, to perform object removal while maintaining semantic consistency across tasks. PixelHacker Xu et al. (2025) modifies latent diffusion by using separate embeddings for foreground and background features. OmniEraser Wei et al. (2025) leverages both the foreground object and background images as independent guidance signals, enhancing the model's contextual understanding.

## 3 METHODOLOGY

### 3.1 OVERALL FRAMEWORK

**Preliminaries.** Given an input image $x$ and a user-defined mask $m$, object removal aims to restore the masked regions $\hat{x}$ with content that is semantically aligned and color-consistent with the surrounding background. Our framework builds upon FLUX.1-dev Black-Forest-Labs (2024), a DiT-based Peebles & Xie (2023b) diffusion model with strong generative capacity. In the forward diffusion process, Gaussian noise is progressively added to a clean image $\hat{x}$ to obtain a noisy sample $x_t$ at timestep $t$: $x_t = \sqrt{\bar{\alpha}_t}, \hat{x} + \sqrt{1 - \bar{\alpha}_t}\epsilon, \epsilon \sim \mathcal{N}(0, I)$, where $\bar{\alpha}t$ denotes the diffusion noise schedule and $\epsilon$ is standard Gaussian noise. In the reverse process, a DiT network $\epsilon_\theta$ predicts the noise $\epsilon_t$ at each timestep, enabling image reconstruction through iterative denoising. The model is trained with the flow-matching objective:

$$\mathcal{L}_{flow} = \mathbb{E}_{x_0,t,\epsilon_t} b(t)^2 \|\epsilon_t - \epsilon_\theta(x_t, t)\|_2^2, \quad b(t) = -\frac{d\bar{\alpha}_t}{dt}\left(\frac{\bar{\alpha}_t}{\sqrt{1-\bar{\alpha}_t}^2} + \frac{\sqrt{1-\bar{\alpha}_t}^2}{\bar{\alpha}_t}\right). \quad (1)$$

To enable object removal, we extend the DiT input layer $\epsilon_\theta$ to accept the masked foreground $x \odot m$, the visible background $x \odot (1 - m)$, the binary mask $m$, and the noisy latent $x_0$. These inputs are concatenated and projected through a learnable linear layer, providing explicit object–background guidance for denoising. Following Wei et al. (2025), the training objective becomes:

$$\mathcal{L}'_{flow} = \mathbb{E}_{x_0,y,t,\epsilon_t} b(t)^2 \|\epsilon_t - \epsilon_\theta(x_t \mid y; t)\|_2^2, \quad (2)$$

where $y = [m, x_0 \odot m, x_0 \odot (1 - m)]$ denotes the conditioning input. This design improves the model's understanding of both masked regions and surrounding context, thereby enhancing background consistency and reducing artifact generation.

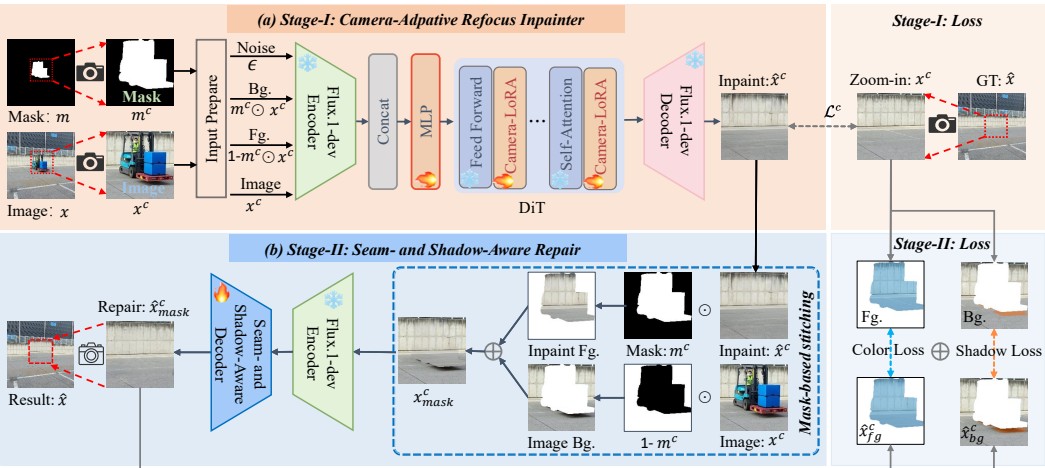

Figure 2: **Pipeline of ReFocusEraser,** including: (1) Stage-I, *Camera-Adpative Refocus Inpainter* enlarges small masked objects via simulated near-field perspectives and improves semantic consistency through LoRA fine-tuning on the enlarged regions, and (2) Stage-II, *Seam- and Shadow-Aware Repair* accepts the zoomed-in image from mask-based stitching and fine-tunes Flux.1-dev decoder to seamlessly integrate inpainted regions and corrects small-object shadows automatically.

**Overview of ReFocusEraser.** While explicit guidance can improve object removal, the fixed compression rate of the VAE encoder imposes an inherent bottleneck for small object restoration, causing high-frequency details to be lost during downsampling Rombach et al. (2022). To address this challenge, we propose ReFocusEraser, a two-stage diffusion-based framework specifically designed for small masked objects. It combines camera-adaptive refocusing to enlarge small objects, LoRA-based fine-tuning of the diffusion model to enhance semantic alignment, and a seam- and shadow-aware decoder with mask-based stitching to preserve background consistency and automatically correct shadows. This design enables detail-preserving, artifact-free removal of small objects while maintaining visually and semantically coherent inpainting results (Figure 2).

## 3.2 CAMERA-ADAPTIVE REFOCUS INPAINTER

**Camera-Adaptive Refocus Mechanism.** Since the sizes of target objects vary across input images $x$, applying a fixed crop or zoom often fails to preserve small masked objects. To address this, we introduce a *camera-adaptive refocus strategy* that dynamically enlarges small masked objects, producing a camera-adaptive input $x^c$ and mask $m^c$ that better preserves fine details during inpainting. Concretely, we first estimate the focal length of the input image $x$ using a camera calibration algorithm Veicht et al. (2024) and classify the scene as far-field or near-field. Based on empirical observation, we set 500 pixels as the focal length threshold, and images with focal lengths below this are considered far-field scenes. Far-field scenes typically correspond to a large field of view, where masked objects are relatively small. Directly compressing such images through the VAE leads to severe information loss. To mitigate this, we apply a camera-conditioned zoom-in to bring the scene closer and relatively enlarge small masked objects. Details are in Appendix A.

**LoRA for DiT-Based Refocus Inpainter.** To improve inpainting of small or distant objects, we replace the original image $x$ with the camera-adaptive zoomed input $x^c$ and apply Low-Rank Adaptation (LoRA) Hu et al. (2022) to the general-purpose Flux-1.dev. The low-rank adapters are injected into all linear layers of the DiT backbone, including self-attention and feedforward blocks. Formally, for a linear layer with weight matrix $W \in \mathbb{R}^{d_{\text{out}} \times d_{\text{in}}}$ and input feature $\mathbf{X} \in \mathbb{R}^{d_{\text{in}}}$ derived from camera-adaptive zooming, the adapted output is $W_{\text{LoRA}}\mathbf{X} = W\mathbf{X} + \frac{\alpha}{r}BA\mathbf{X}$, where $A \in \mathbb{R}^{r \times d_{\text{in}}}$ and $B \in \mathbb{R}^{d_{\text{out}} \times r}$ are trainable low-rank matrices, $r \ll \min(d_{\text{in}}, d_{\text{out}})$ is the rank, and $\alpha$ is a scaling factor. This enables the model to effectively leverage zoom-adaptive features throughout the generative

process by minimizing the flow-based loss:

$$\mathcal{L}_{flow}^c = \mathbb{E}_{\boldsymbol{x}_0^c, \boldsymbol{y}^c, t, \epsilon_t} b(t)^2 \|\epsilon_t - \epsilon_\theta(\boldsymbol{x}_t^c \mid \boldsymbol{y}^c; t)\|_2^2, \tag{3}$$

where $\boldsymbol{y}^c = [\boldsymbol{m}^c, \boldsymbol{x}_0^c \odot \boldsymbol{m}^c, \boldsymbol{x}_0^c \odot (1 - \boldsymbol{m}^c)]$. Unlike the inpainting-specialized Flux-Fill Black-Forest-Labs (2024) and Flux-Kontext Batifol et al. (2025), *applying LoRA to the general-purpose Flux-1.dev preserves pre-trained parameters while enhancing cross-task transferability.*

### 3.3 SEAM- AND SHADOW-AWARE REPAIR

**Mask-based Stitching Strategy.** To reintegrate the inpainted patch $\hat{\boldsymbol{x}}^c$ into the original image $\boldsymbol{x}$, we consider two stitching strategies (see visual comparison in Appendix B). A straightforward option is *box-based stitching*, which scales and pastes the inpainted patch back into the original image: $\boldsymbol{x}^{\text{box}} = \boldsymbol{x} + S(\hat{\boldsymbol{x}}^c)$. While this strategy preserves the full patch structure, it often introduces boundary misalignment and visible seams. In contrast, *mask-based stitching* pastes only the inpainted masked region: $\boldsymbol{x}^{\text{mask}} = \boldsymbol{x} + S((1 - \boldsymbol{m}^c) \odot \boldsymbol{x}^c + \boldsymbol{m}^c \odot \hat{\boldsymbol{x}}^c)$, thereby maintaining the surrounding context unchanged. Despite minor color inconsistencies near the mask boundary, mask-based stitching ensures better spatial alignment and semantic coherence, making it the preferred strategy in our framework. *Notably, mask-based stitching cannot remove shadows cast by the unwanted object*, which may degrade inpainting quality or introduce spatial inconsistencies.

**Seam- and Shadow-Aware Repair.** To correct color shifts and automatically remove shadows introduced by mask-based stitching, we propose a *Seam- and Shadow-Aware Repair* stage that leverages the strong reconstruction capability of the FLUX.1-dev VAE Black-Forest-Labs (2024) to restore contextual consistency. This is achieved by fine-tuning only the decoder, as fine-tuning the entire VAE often results in severe catastrophic forgetting of the pretrained reconstruction prior. Addressing this issue would require substantially larger training datasets, longer optimization schedules, and significantly higher computational costs Otani (2025), which are impractical given our current data volume and computational resources. By fine-tuning only the decoder, we maximally leverage the pretrained FLUX.1-dev VAE's reconstruction abilities, reduce model complexity, and enable Stage-II to focus exclusively on correcting color shifts and shadow artifacts without disrupting the global scene representation learned in Stage-I. Let $\hat{\boldsymbol{x}}^c$ denote the inpainted result from Stage-I, and the mask-stitched input be:

$$\boldsymbol{x}_{\text{mask}}^c = (1 - \boldsymbol{m}^c) \odot \boldsymbol{x}^c + \boldsymbol{m}^c \odot \hat{\boldsymbol{x}}^c. \tag{4}$$

We initialize the Stage-II decoder $\mathcal{D}_{\text{seam}}$ from the Stage-I decoder and fine-tune it to specifically correct color discrepancies and mitigate shadow artifacts at mask boundaries.

Since finetuning VAE with L1 or L2 reconstruction losses often produces blurry outputs, particularly for small or distant objects. This is because downscaling and encoding compresses fine-grained information in the latent space, causing the decoder to lose high-frequency details such as edges, textures, and shadows Rombach et al. (2022). When the encoder is frozen, the decoder cannot rely on updated latent representations to recover lost details, making simple L1/L2 supervision insufficient.

**Color-Shadow Consistency Loss.** To effectively fine-tune the Stage-II decoder, we introduce a color-shadow consistency loss $\mathcal{L}_{\text{color\_shadow}}$ that separately supervises the inpainted foreground and surrounding background. Specifically, an $L_1$ loss is applied to the masked foreground to maintain pixel-level color consistency, while LPIPS Zhang et al. (2018b) is applied to the unmasked background to preserve high-frequency details, including shadows. Formally, the loss is defined as

$$\mathcal{L}_{\text{color\_shadow}} = \mathbb{E}\left[\|\boldsymbol{x}_{\text{fg}}^c - \hat{\boldsymbol{x}}_{\text{fg}}^c\|_1\right] + \mathbb{E}\left[\text{LPIPS}(\boldsymbol{x}_{\text{bg}}^c, \hat{\boldsymbol{x}}_{\text{bg}}^c)\right], \tag{5}$$

where $\boldsymbol{x}_{\text{fg}}^c$ and $\boldsymbol{x}_{\text{bg}}^c$ denote the masked foreground and background regions from the mask-based stitching $\boldsymbol{x}_{\text{mask}}^c$, respectively. This region-aware perceptual supervision enables the decoder to restore fine object structures while maintaining realistic and consistent background. Finally, the refined inpainted region is mapped back to the original image through a camera-aligned transformation, ensuring seamless spatial and color consistency in the final output.

## 4 EXPERIMENTS

### 4.1 EXPERIMENTAL SETUPS

**Datasets.** We train our models on 12,757 high-resolution image pairs from the Real-world Object Removal Dataset (RORD) Sagong et al. (2022), where each input consists of an image and a corresponding mask indicating the unwanted object, and evaluate on 1,542 RORD validation pairs as well as the real-world benchmark RemovalBench Wei et al. (2025), ensuring both controlled and in-the-wild performance assessments. See Appendix C for data collection and preprocessing details.

**Evaluation Metrics.** We evaluate our method from two complementary perspectives: (1) overall image quality, assessed by FID Heusel et al. (2017) and CMMD Jayasumana et al. (2024), which capture distributional alignment and semantic similarity to ground truth; and (2) region-level fidelity, measured by PSNR Hore & Ziou (2010), SSIM Wang et al. (2004), and LPIPS Zhang et al. (2018b), which reflect pixel accuracy and perceptual consistency. *To ensure fair comparison, all evaluation metrics are computed on the full image after reinserting the zoomed-in inpainted patch.*

**Implementation Details.** Our framework is built on the general-purpose FLUX.1-dev and trained in two stages. In Stage I, we apply LoRA as a plug-and-play adaptation, trained with rank 32, a learning rate of 3e-5, and batch size 2 for 20 epochs on 4 NVIDIA H200 GPUs ($\approx$ 2 days). In Stage II, we fine-tune a dedicated VAE decoder to correct color and shadow inconsistencies, while reusing the pretrained encoder from FLUX.1-dev. The decoder is optimized with a learning rate of 1e-4 and batch size 4 for up to 80K steps on 8 H200 GPUs ($\approx$ 1.5 days).

**Baseline and Comparisons.** We adopt OmniEraser Wei et al. (2025) as our baseline and train it on the RORD dataset. To demonstrate the superiority of our proposed method, we compare it with state-of-the-art diffusion-based approaches for both general inpainting (RePaint Lugmayr et al. (2022), PixelHacker Xu et al. (2025), Flux-Knotext Batifol et al. (2025), Flux-Fill Black-Forest-Labs (2024)) and object removal (AttentiveEraser Sun et al. (2025), CLIPAway Ekin et al. (2024), PowerPaint Zhuang et al. (2024), OmniEraser Wei et al. (2025)). All results are evaluated using their official implementations and default settings to ensure a fair and consistent comparison.

### 4.2 COMPARISON WITH STATE-OF-THE-ARTS

**Quantitative Comparisons.** Table 1 reports our method's improvements over state-of-the-art approaches in overall image quality and region-level fidelity. **1) For region-level fidelity**, Flux-Fill, which takes a text description and binary mask, achieves the prior best results (PSNR 25.025, SSIM 0.794, LPIPS 0.092) on RORD-val for small-object removal. Compared to this, our ReFocusEraser substantially improves reconstruction quality, *raising PSNR to 31.256 (+6.231), SSIM to 0.924 (+0.13), and lowering LPIPS to 0.041 (–0.051).* Similarly, on RemovalBench for normal-object removal, AttentiveEraser outperforms previous methods, whereas ReFocusEraser further surpasses it, achieving PSNR 30.495, SSIM 0.841, and LPIPS 0.223, compared to 25.365, 0.751, and 0.284, respectively. **2) For overall image quality**, AttentiveEraser achieves the previous best results on both RORD and RemovalBench. Our ReFocusEraser further improves performance, achieving lower FID and CMMD scores on both benchmarks (*e.g., FID* 25.393 $\rightarrow$ 21.378 *and CMMD* 0.276 $\rightarrow$ 0.263 *on RORD, FID* 65.326 $\rightarrow$ 38.115 *and CMMD* 0.169 $\rightarrow$ 0.117 *on RemovalBench*). These results show that our method excels at maintaining detail and semantic consistency across diverse scenes.

**Qualitative Comparisons.** Figure 3 visually compares our method with prior methods, highlighting its superior object removal and region repair. From the visual results in columns (b)–(e), general inpainting methods struggle with object removal; even when Flux-Kontext and Flux-Fill succeed, they often fail to restore the removed regions consistently with the background (*see green boxes in the first row*). From the results in columns (f)–(i), early specialized removal methods such as Power-Paint and CLIPAway fail to effectively remove and repair target regions. While AttentiveEraser and

Table 1: **Quantitative comparison of our ReFocusEraser with state-of-the-arts.**

| Method | RORD-val Sagong et al. (2022) | | | | | RemovalBench Wei et al. (2025) | | | | |
|---|---|---|---|---|---|---|---|---|---|---|
| | PSNR↑ | SSIM↑ | LPIPS↓ | FID↓ | CMMD↓ | PSNR↑ | SSIM↑ | LPIPS↓ | FID↓ | CMMD↓ |
| Repaint Lugmayr et al. (2022) | 15.250 | 0.330 | 0.787 | 144.320 | 1.011 | 20.747 | 0.705 | 0.564 | 155.695 | 0.548 |
| PixelHacker Xu et al. (2025) | 22.600 | 0.530 | 0.340 | 44.108 | 0.499 | 22.058 | 0.728 | 0.425 | 143.315 | 0.483 |
| CLIPAway Ekin et al. (2024) | 21.916 | 0.511 | 0.333 | 61.670 | 0.535 | 22.076 | 0.721 | 0.412 | 140.610 | 0.404 |
| PowerPaint Zhuang et al. (2024) | 23.229 | 0.619 | 0.176 | 42.519 | 0.301 | 23.228 | 0.732 | 0.323 | 143.619 | 0.463 |
| OmniEraser Wei et al. (2025) | 22.380 | 0.562 | 0.226 | 47.765 | 0.293 | 24.624 | 0.728 | 0.291 | 66.871 | 0.324 |
| Flux-Knotext Batifol et al. (2025) | 13.522 | 0.251 | 0.628 | 127.330 | 0.824 | 22.234 | 0.679 | 0.317 | 88.768 | 0.191 |
| Flux-Fill Black-Forest-Labs (2024) | 25.025 | 0.794 | 0.092 | 65.215 | 0.297 | 21.526 | 0.744 | 0.348 | 177.571 | 0.434 |
| AttentiveEraser Sun et al. (2025) | 24.064 | 0.656 | 0.145 | 25.393 | 0.276 | 25.265 | 0.751 | 0.284 | 65.326 | 0.169 |
| **Our ReFocusEraser** | **31.256** | **0.924** | **0.041** | **21.378** | **0.263** | **30.495** | **0.841** | **0.223** | **38.115** | **0.117** |

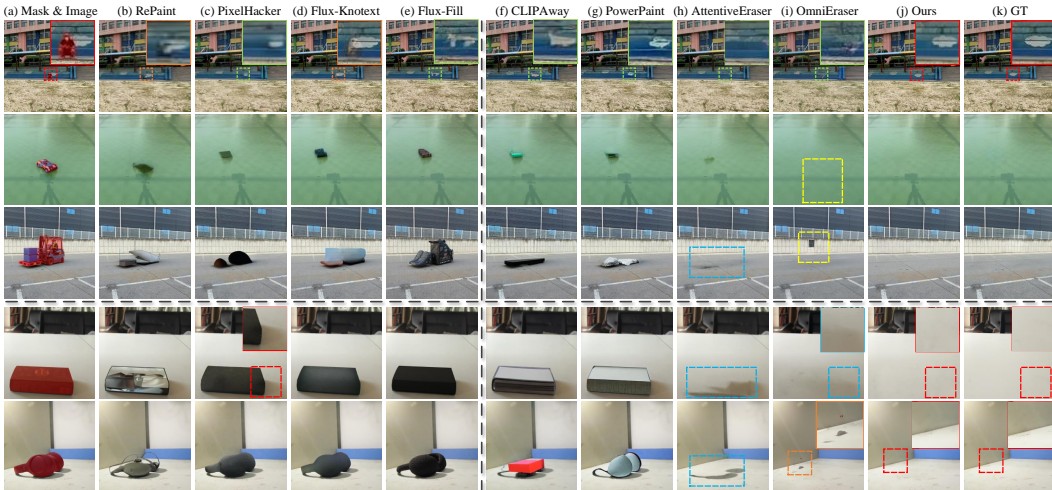

Figure 3: **Qualitative comparison of our method with state-of-the-arts** on both the RORD-val (top three rows) and RemovalBench (bottom two rows), including general inpainting methods (b) **RePaint**, (c) **PixelHacker**, (d) **Flux-Fill**, (e) **Flux-Knotext** and specialized object removals (f) **CLIPAway**, (g) **PowerPaint**, (h) **AttentiveEraser**, (i) **OmniEraser**, and (j) our **ReFocusEraser**.

OmniEraser can remove objects, they often fail to consistently repair the background. As highlighted by the yellow boxes in column (i), OmniEraser removes the foreground but disrupts background consistency. AttentiveEraser preserves background consistency but leaves object shadows. In contrast, our ReFocusEraser consistently removes objects while maintaining structural and semantic consistency.

## 4.3 ANALYSIS OF BASELINE IMPROVEMENTS

Table 2 and Figure 4 summarize the improvements from our proposed components: Camera-Adaptive Refocus (CAR), mask-based stitching, and Seam- and Shadow-Aware Repair. As highlighted by the yellow boxes in columns (a) and (e) of Figure 4, the baseline trained via LoRA on the original images can remove unwanted objects but often disrupts background consistency due to asymmetry between the VAE encoder and decoder. Incorporating the CAR mechanism substantially improves performance: *compared to (a) in Table 2, (b) raises PSNR by 7.399 and SSIM by 0.297, while lowering LPIPS, FID, and CMMD by 0.277, 15.999, and 0.218, respectively.* Mask-based stitching reduces the severe color shifts of box-based stitching and further improves region-level fidelity (PSNR +0.65, SSIM +0.009, LPIPS –0.007), although FID and CMMD slightly degrade due to remaining shadows. As shown in Figures 4 (c)–(d), introducing the Seam- and Shadow-Aware Repair effectively corrects color seams while automatically removing object shadows.

Table 2: **Quantitative evaluation against baseline.** 'CAR' denotes the Camera-Adaptive Refocus mechanism. '$S_{mask}$' and '$S_{box}$' refer to mask-based and box-based stitching strategies. (a) is the baseline trained on the original images, while (b)–(d) are trained on $3\times$ zoomed-in images.

| Exp. | Configuration | | | Capabilities | | | RORD-val Sagong et al. (2022) | | | | |
| | LoRA | Stitch | Decoder | Seam | Shadow | Color Shift | PSNR↑ | SSIM↑ | LPIPS↓ | FID↓ | CMMD↓ |
|---|---|---|---|---|---|---|---|---|---|---|---|
| (a) | ✗ | ✗ | ✗ | N | Y | N | 23.683 | 0.626 | 0.323 | 40.287 | 0.476 |
| (b) | CAR | $S_{box}$ | ✗ | Y | N | Y | 31.082 | 0.923 | 0.046 | 24.288 | **0.258** |
| (c) | CAR | $S_{mask}$ | ✗ | Y | Y | Y | **31.732** | **0.932** | **0.039** | 25.036 | 0.272 |
| (d) | CAR | $S_{mask}$ | ✓ | N | N | N | 31.256 | 0.924 | 0.041 | **21.378** | 0.263 |

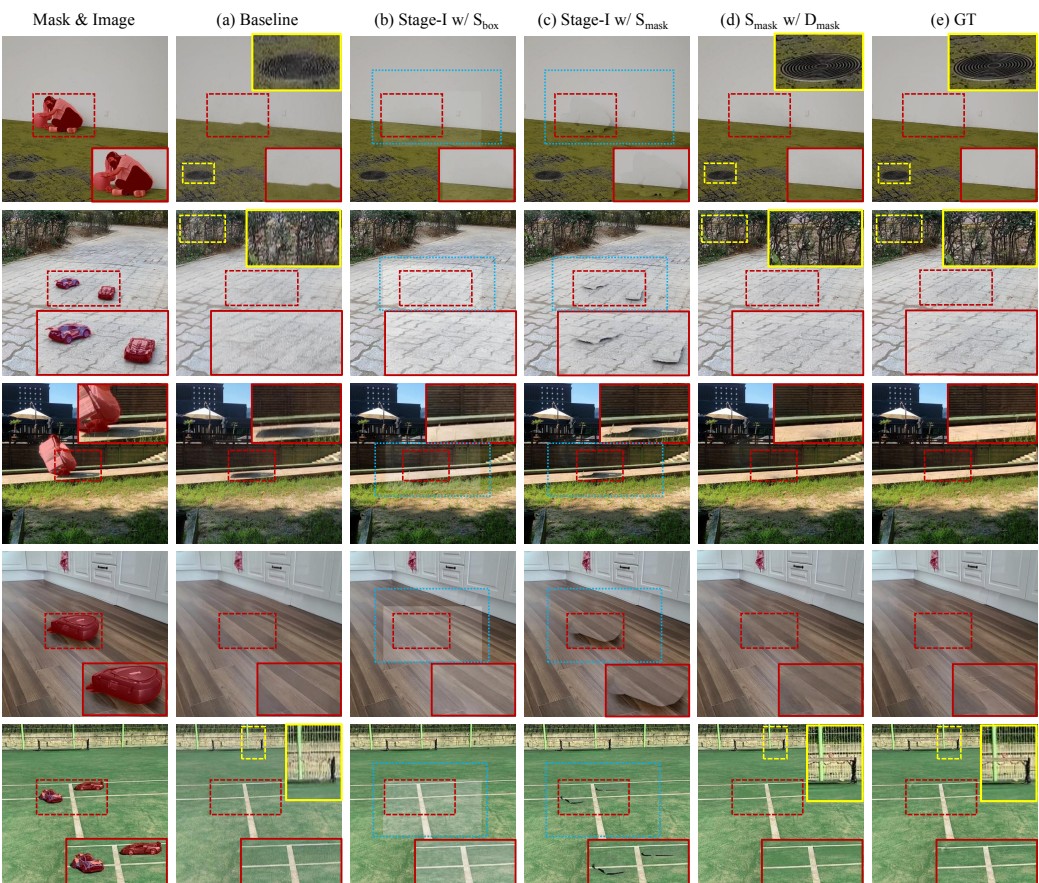

Figure 4: **Visual comparison against baselines**. (a) **Baseline**: trained on original images; (b) **Stage-I w/ $S_{box}$**: trained on $3\times$ images, patch reintegrated via box-based stitching; (c) **Stage-I w/ $S_{mask}$**: trained on $3\times$ images, patch reintegrated via mask-based stitching. (d) $S_{mask}$ **w/ $D_{mask}$**: Stage-II Seam- and Shadow-Aware Decoder, added on top of (c) to repair seam and shadow.

## 4.4 ABLATION STUDIES

We perform ablations on ReFocusEraser, analyzing the Camera-Adaptive Refocus Inpainter, stitching strategies, and loss functions for Seam- and Shadow-Aware Repair.

**Effect of Camera-Adaptive Refocus Inpainter under Different Focal Lengths.** Table 3 shows that increasing focal length consistently improves inpainting performance. Compared to $1\times$, training at $2\times$ improves PSNR by 2.856 and SSIM by 0.198, while reducing LPIPS by 0.184, FID by 20.69, and CMMD by 0.43. Extending to $3\times$ yields further gains, with PSNR +5.891, SSIM +0.265,

Table 3: **Performance on camera-adaptive refocus inpainter under various focal lengths.**

| #Focal Length | RORD-val Sagong et al. (2022) | | | | |
|---|---|---|---|---|---|
| | PSNR ↑ | SSIM ↑ | LPIPS ↓ | FID ↓ | CMMD ↓ |
| 1× | 23.683 | 0.626 | 0.323 | 40.287 | 0.476 |
| 2× | 26.539 (+2.856) | 0.824 (+0.198) | 0.139 (-0.184) | 19.597 (-20.690) | 0.046 (-0.43) |
| 3× | **29.574** (+5.891) | **0.891** (+0.265) | **0.096** (-0.227) | **13.283** (-27.004) | **0.026** (-0.45) |

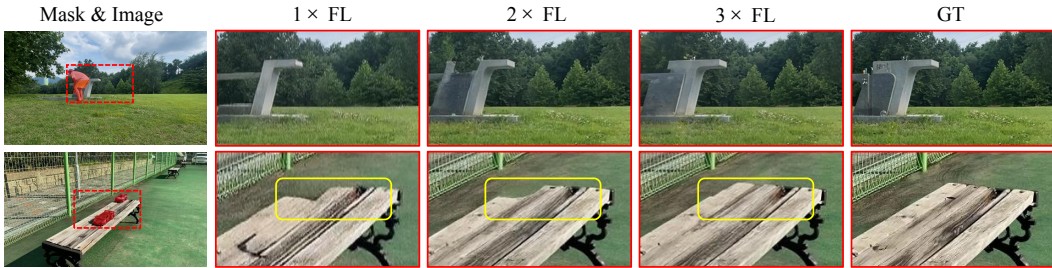

Figure 5: **Visual comparison of camera-adaptive refocus inpainter under various focal lengths.**

and reductions of 0.227, 27.004, and 0.45 in LPIPS, FID, and CMMD, respectively. Visual results in Figure 5 (yellow boxes) confirm that higher focal lengths recover finer semantic details. This improvement arises because, under the fixed compression ratio of the inpainting network, increasing the focal length allows the model to perceive finer details, effectively providing a more suitable compression of the input image. Training at 3× yields the best trade-off, highlighting the effectiveness of Camera-Adaptive Refocus in improving both perceptual quality and reconstruction fidelity.

**Mask-based vs. Box-based Stitching Methods.** Table 4 compares mask- and box-based stitching for reinserting zoomed-in inpainted patches into the original image using the inverse camera calibration algorithm Veicht et al. (2024). Mask-based stitching consistently improves region-level fidelity over box-based stitching, achieving higher PSNR and SSIM and lower LPIPS at both 2× (PSNR +0.66, SSIM +0.012, LPIPS –0.013) and 3× (PSNR +0.65, SSIM +0.009, LPIPS –0.007). For overall image quality, CMMD is comparable (0.279 vs. 0.270 at 2×, 0.272 vs. 0.258 at 3×), though FID is slightly higher (13.041 vs. 8.458 at 2×; 25.036 vs. 24.288 at 3×) due to shadow inconsistencies during mask-based reinsertion. As shown in Figure 6 (a) and (c), smaller focal lengths in Stage-I produce more noticeable box seams. In contrast, mask-based stitching introduces seam artifacts only along object boundaries, which are fixed, primarily affecting de-

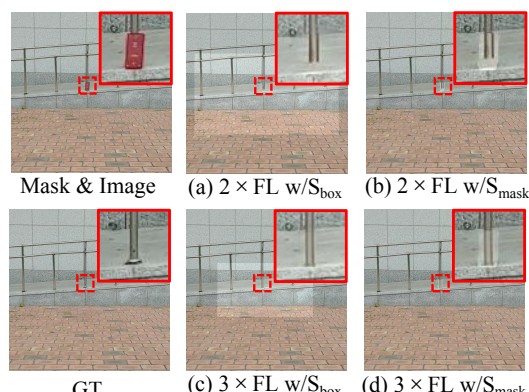

Figure 6: **Visual comparison of mask- vs. box-based stitching.** '× FL w/ **S**' denotes Stage-I inpainted result at 2× and 3× focal lengths pasted into the input image using different stitchings.

tails within the masked region (see Figure 6 (b) and (d)). These results suggest that mask-based stitching better preserves background integrity and visual coherence, making it a more effective strategy for unwanted object removal.

**Fine-tune Flux.1-dev Decoder at Stage-II with different losses.** Our seam- and shadow-aware repair builds on the VAE of FLUX.1-dev, with only the decoder fine-tuned. Table 5 and Figure 7 evaluate the effectiveness of our proposed color–shadow consistency loss, which applies LPIPS to the background and L1 to the foreground to address both color shift and shadow artifacts. Observing

Table 4: **Performance of different stitching strategies for Stage-I inpainted results at 2× and 3× focal lengths.** Mask-based stitching consistently outperforms box-based stitching.

| #Stitching | RORD-val Sagong et al. (2022) | | | | |
|---|---|---|---|---|---|
| | PSNR ↑ | SSIM ↑ | LPIPS ↓ | FID ↓ | CMMD ↓ |
| 2× FL w/ $\mathbf{S}_{box}$ | 31.245 | 0.919 | 0.053 | 8.458 | 0.270 |
| 2× FL w/ $\mathbf{S}_{mask}$ | **31.905** (+0.66) | **0.931** (+0.012) | **0.040** (-0.013) | 13.041 (+4.583) | 0.279 (+0.009) |
| 3× FL w/ $\mathbf{S}_{box}$ | 31.082 | 0.923 | 0.046 | 24.288 | 0.258 |
| 3× FL w/ $\mathbf{S}_{mask}$ | **31.732** (+0.65) | **0.932** (+0.009) | **0.039** (-0.007) | 25.036 (+0.748) | 0.272 (+0.014) |

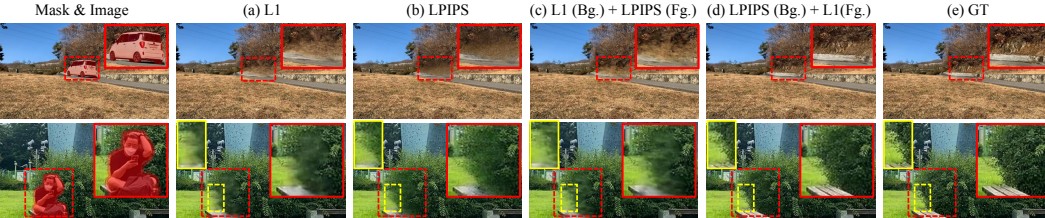

Figure 7: **Visual comparison of fine-tuning the Flux.1-dev decoder with different losses.** Using LPIPS for the background and L1 for the foreground not only ensures visual consistency to address seam issues caused by color shifts but also effectively repairs object shadows.

Table 5: **Comparison of fine-tuning the Flux.1-dev decoder with different losses.**

| #Loss | RORD-val Sagong et al. (2022) | | | | |
|---|---|---|---|---|---|
| | PSNR ↑ | SSIM ↑ | LPIPS ↓ | FID ↓ | CMMD ↓ |
| L1 | 30.954 | 0.922 | 0.044 | 24.580 | 0.275 |
| LPIPS | 30.920 (-0.034) | 0.921 (-0.001) | 0.043 (-0.001) | 24.618 (+0.038) | **0.262** (-0.013) |
| L1(Bg.)+LPIPS(Fg.) | 30.948 (-0.006) | 0.921 (-0.001) | 0.044 | 24.124 (-0.456) | 0.283 (+0.008) |
| LPIPS(Bg.)+L1(Fg.) | **31.256** (+0.302) | **0.924** (+0.002) | **0.041** (-0.003) | **21.379** (-3.201) | 0.263 (-0.012) |

Figure 7 (a) and (b), training with only L1 loss limits foreground repair and causes background detail loss, while using only LPIPS preserves background semantics but fails to restore fine foreground regions. As reported in Table 5, both settings yield nearly identical scores (e.g., PSNR 30.954 vs. 30.920, SSIM 0.922 vs. 0.921). This reflects their complementary strengths: L1 enforces pixel-level accuracy, whereas LPIPS emphasizes perceptual similarity but is less sensitive to small objects. Based on this observation, we explored combining LPIPS and L1 losses to jointly train the decoder. Using LPIPS for foreground repair with L1 for background preservation yields only a modest FID improvement (–0.456 in Table 5), while other metrics remain largely unchanged, consistent with the visual results in Figure 7 (a) and (c). In contrast, applying LPIPS to background reconstruction with L1 regularization on the foreground (last row) achieves the best overall performance, providing superior metric gains and improved visual quality. As shown in Figure 7 (d), this setting ensures seamless color consistency between the inpainted region and surrounding context and effectively removes residual shadows, demonstrating the effectiveness of our color-shadow consistency loss.

## 5 CONCLUSIONS

In this paper, we address small object removal, a task where existing methods often struggle to accurately erase and repair fine details. We propose RefocusEraser, a two-stage framework that combines small object refocusing with robust context- and shadow-aware repair. In Stage I, a Camera-Adaptive Refocus Inpainter enlarges masked objects and enhances semantic consistency through LoRA fine-tuning. Stage II introduces Seam- and Shadow-Aware Repair with mask-based stitching to seamlessly integrate inpainted regions and remove shadows. Extensive experiments on benchmark datasets demonstrate the effectiveness of our approach, and the framework generalizes to other tasks requiring cross-scale semantic alignment and seamless region integration.

ACKNOWLEDGEMENTS

This work is supported by the National Natural Science Foundation of China (NSFC) 62502448.

## ETHICS STATEMENT

Our work introduces RefocusEraser, a diffusion-based framework for object removal. While our method can benefit applications such as image editing, content restoration, and virtual background generation, it could also be misused to generate highly realistic synthetic images, potentially raising ethical concerns regarding privacy, consent, and misinformation.

To mitigate these risks, all experiments are conducted using publicly available datasets with appropriate licenses, and no personally identifiable or sensitive information is included. We encourage responsible use of our models and advise careful consideration of ethical and societal implications when deploying such generative technologies. Our primary goal is to advance scientific research in controllable image generation, and we advocate for precautions to prevent malicious or unethical applications.

## USE OF LARGE LANGUAGE MODELS STATEMENT

We did not use any large language models (LLMs) in the development, experimentation, or writing of this work. All models, algorithms, and analyses presented in this paper were implemented and evaluated without reliance on LLMs. The datasets, training procedures, and evaluation protocols are entirely independent of LLM-generated content. This ensures that our results and conclusions are fully derived from the methods described in this paper and are not influenced by any external LLM outputs.

## REPRODUCIBILITY STATEMENT

We provide a detailed description of our RefocusEraser framework, which is optimized through two decoupled training stages: Stage-I Camera-Adaptive Refocus Inpainter and Stage-II Seam- and Shadow-Aware Repair. Each stage is fine-tuned separately to address its specific objective. We include comprehensive training details, hyperparameter settings, and evaluation protocols in both the main paper and supplementary material to facilitate replication. All datasets used are publicly available with appropriate licenses. While the full source code will be released upon publication, all results reported in this work can be reproduced using the provided descriptions and publicly accessible data.

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

## A    DETAILS OF CAMERA-ADAPTIVE REFOCUS STRATEGY

***Step 1:*** **Focal-Length-Aware Image Selection.**    Since the original images lack known focal lengths, estimating camera parameters is essential for our refocus mechanism. We first predict the camera intrinsics—including roll, pitch, vertical field of view (vFoV), and focal length—using GeoCalib Veicht et al. (2024). Then, using images captured by real cameras as a reference, we align the predicted focal lengths with ground-truth parameters. Based on these aligned values, we filter and select images for our dataset.

***Step 2:*** **Camera-Adaptive Refocus Strategy.**    After obtaining the images based on their estimated focal lengths, we introduce a Camera-Adaptive Refocus Strategy, which leverages camera-adaptive zoom-in to simulate varying focal perspectives. Specifically, we design a transformation pipeline that performs central field-of-view (FoV) cropping and rescaling to generate images at zoom factors of $2\times$, $3\times$, and $4\times$, corresponding to different effective focal lengths. Given an original image of size $\mathbf{W} \times \mathbf{H}$ and a target zoom factor $z \in 2, 3, 4$, we define the cropping ratio as $r = 1/z$. Using this ratio, we extract a central region of width $\mathbf{W}' = r\mathbf{W}$ and height $\mathbf{H}' = r\mathbf{H}$ to simulate the visual narrowing effect of increased focal length. Formally, the cropping operation can be expressed as:

$$\boldsymbol{x}_{zoom} = \text{Resize}\left(\boldsymbol{x}\left[\tfrac{(1-r)}{2}\mathbf{W} : \tfrac{(1+r)}{2}\mathbf{W}, \tfrac{(1-r)}{2}\mathbf{H} : \tfrac{(1+r)}{2}\mathbf{H}\right]\right) \tag{6}$$

where $\boldsymbol{x}_{\text{zoom}}$ denotes the image generated at a different focal length. The resizing operation produces images with distinct focal characteristics, corresponding to zoom factors of $2\times$, $3\times$, and $4\times$.

***Step 3:*** **Complete-Mask Image Filter.**    To prevent partial truncation of foreground objects during the zoom-in cropping process, particularly at higher focal lengths, we ensure foreground completeness through a mask-based image filtering strategy. Let $\sum \boldsymbol{m}'$ denote the number of foreground pixels in the original mask $\boldsymbol{m}$, and let $\boldsymbol{m}_{z\times}$ represent its corresponding zoomed-in version at zoom level $z \in 2, 3, 4$. To guarantee foreground completeness, we compare the actual foreground area in $\boldsymbol{m}_{z\times}$ with the expected area under ideal central cropping. Assuming the foreground is fully contained within the central crop, the expected area after zooming is $\frac{1}{z^2} \sum \boldsymbol{m}'$ of the original mask area. We define the completeness error rate $\delta$ as:

$$\delta = \frac{\left|\sum \boldsymbol{m}_{z\times} - \frac{1}{z^2} \sum \boldsymbol{m}'\right|}{\frac{1}{z^2} \sum \boldsymbol{m}'} \tag{7}$$

where $\sum \boldsymbol{m}_{z\times}$ denotes the number of foreground pixels in the zoomed-in mask. A sample is retained only if the completeness error $\delta$ is below a predefined threshold $\tau$ (set to 0.1); otherwise, it

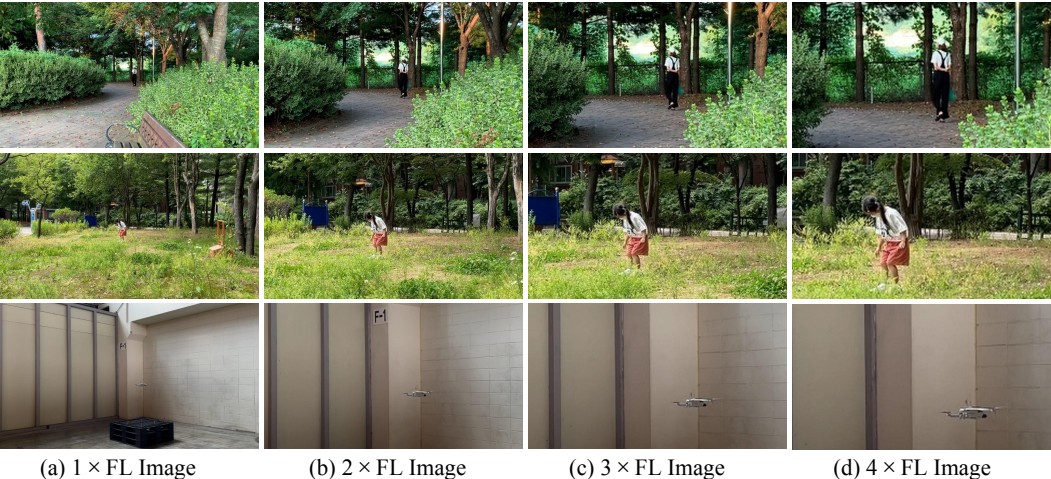

|  |  |  |  |
|---|---|---|---|
| (a) $1 \times$ FL Image | (b) $2 \times$ FL Image | (c) $3 \times$ FL Image | (d) $4 \times$ FL Image |

Figure 8: **Visual examples of simulated camera-adaptive zoom-in images,** where larger focal lengths correspond to wider fields of view, providing the VAE with more contextual information and enabling better preservation of fine structural details.

is discarded. This filtering is applied across zoom levels 2×, 3×, and 4×, resulting in a curated set of zoomed images with verified foreground integrity. Examples of the simulated camera-adaptive zoom-in images are shown in Figure 8, where successive zoom levels replicate narrower fields of view and enable better preservation of fine structural details in the foreground objects.

# B    COMPARISON OF MASK-BASED AND BOX-BASED STITCHING

After obtaining the inpainted outputs from Stage-I, the next challenging step is to reintegrate them into the original image. To analyze the impact of different compositing strategies, we consider two stitching methods, as illustrated in Figure 9:

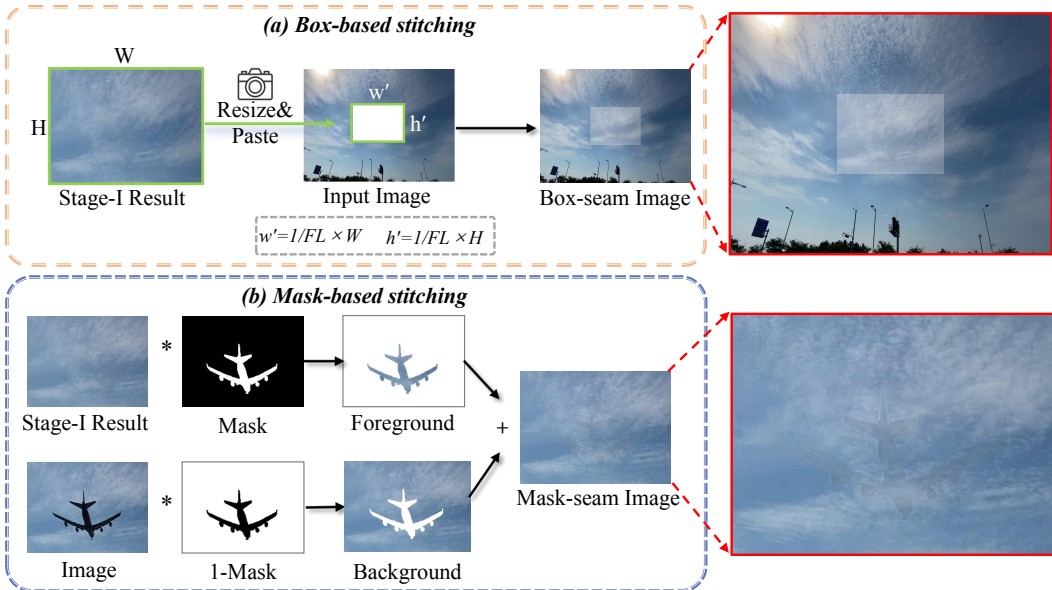

Figure 9: **Performance comparison of stitching strategies.** (a) Box-based stitching introduces visible seams and color inconsistencies along object boundaries, whereas (b) our proposed mask-based stitching achieves better semantic alignment with fewer artifacts.

**Box-based stitching.** A straightforward strategy is to directly resize the inpainted patch and paste it back into the original image according to the zooming scale. However, this box-based compositing introduces noticeable artifacts. Since the entire rectangular patch, including unmodified background regions, is reinserted, sharp seams often emerge along its boundaries. Moreover, local inconsistencies in texture or illumination across the cropped edges disrupt spatial continuity, and these artifacts are further amplified at higher zoom levels.

**Mask-based stitching.** Rather than reinserting the entire rectangular crop, the restored foreground is selectively extracted using the binary mask and blended with the corresponding background. This ensures that only the inpainted pixels are composited back, eliminating redundant overlaps and reducing visible seams along object boundaries, thus improving spatial continuity and visual coherence. Nevertheless, since the inpainted region is produced by the VAE decoder, subtle color discrepancies may still arise between the restored foreground and the untouched background, occasionally resulting in faint mask-shaped seams at the boundaries.

# C    DATA COLLECTION AND CONSTRUCTION

**Data Collection.** To train both the LoRA module and the VAE decoder in ReFocusEraser, we construct a high-quality dataset for small object removal, consisting of paired foreground images and their corresponding ground-truth backgrounds. This dataset is derived from RORD, a large-scale real-world object removal dataset containing 516,709 high-resolution images (960×540) across 3,447 unique scenes. For each scene, we provide both a foreground image and a background image

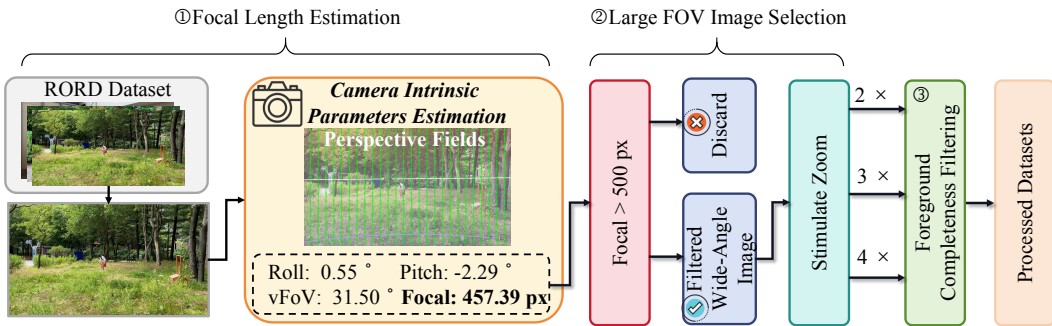

Figure 10: **Overview of our data preprocessing pipeline,** including: ① **Focal Length Estimation**, where camera parameters for all images in the raw RORD dataset are estimated using GeoCalib; ② **Large FOV Image Selection**, which retains images with focal lengths smaller than 500 px; and ③ **Foreground Completeness Filtering**, ensuring mask objects remain complete across all scales.

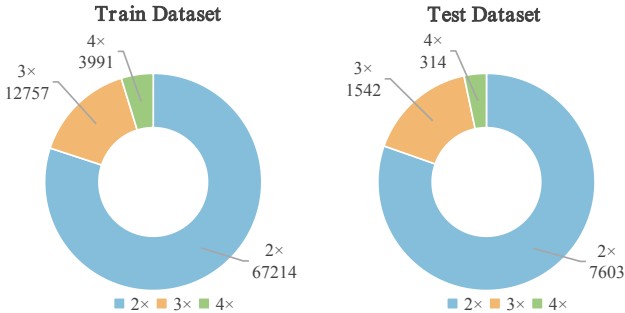

Figure 11: **Train and Test Splits of the Processed RORD Dataset.** The numbers of valid images at each zoom level ($2\times$, $3\times$, $4\times$) for training and testing are listed.

with the target object removed, enabling precise supervision for training. For our experiments, we select a subset of images corresponding to the $2\times$ focal length, comprising 12,757 training images and 1,542 testing images. We employ the object mask annotations provided by RORD, derived from semantic segmentation maps, directly as object masks. When multiple masks correspond to distinct regions of individual objects, we merge them into unified masks to ensure accurate and complete object representations.

**Construction Pipeline.** As shown in Figure 10, our data construction pipeline for RORD consists of three main steps: ① Focal Length Estimation (detailed in Appendix A, Steps 1–2), ② Large FOV Image Selection, and ③ Foreground Completeness Filtering (detailed in Appendix A, Step 3). After applying this pipeline, the selected images are partitioned into training and testing subsets according to their zoom levels. The statistics are summarized in Figure 11. The training set contains 67,214 images at $2\times$, 12,757 images at $3\times$, and 3,991 images at $4\times$, while the testing set includes 7,630 images at $2\times$, 1,542 images at $3\times$, and 314 images at $4\times$, ensuring a balanced distribution across focal lengths for model evaluation.

## D  TRAINING DETAILS

Our ReFocusEraser framework is optimized through two decoupled training stages: (1) **Stage-I Camera-Adaptive Refocus Inpainter** and (2) **Stage-II Seam- and Shadow-Aware Repair**. Each stage is fine-tuned separately to focus on its specific objective. In this section, we elaborate on the corresponding training strategies for both stages.

**Stage-I: SFT for Camera-Adaptive Refocus Inpainter.** In this stage, we build the Camera-Adaptive Refocus Inpainter upon Flux.1-dev Black-Forest-Labs (2024), further augmented with a refocus mechanism. We first train a LoRA module to adapt the model for inpainting across varying focal lengths. Concretely, we freeze all pre-trained DiT parameters and insert lightweight LoRA

layers into both self-attention and feed-forward blocks. Input images and masks are processed with our camera-control strategy, which simulates $n\times$ zoom to enrich contextual detail. The zoomed foregrounds, backgrounds, masks, and noisy inputs are concatenated and linearly projected before being fed into DiT for denoising. Training is supervised via a diffusion loss between the denoised outputs and the ground-truth targets. Owing to the scarcity of high-zoom data, we train independent models for $1\times$, $2\times$, and $3\times$ zoom levels. Each model is optimized for 20 epochs with a LoRA rank of 32, a learning rate of $3e-5$, and a batch size of 2 on four NVIDIA H200 GPUs, requiring approximately two days per run.

**Stage-II: SFT for Seam- and Shadow-Aware Repair.** This stage is built upon the VAE of Flux.1-dev Black-Forest-Labs (2024). The inpainted results from Stage-I are stitched into the corresponding zoom-in inputs using a mask-based compositing strategy, which preserves local structures while avoiding boundary artifacts. The fused outputs are then refined by the VAE to ensure seamless blending and automatic shadow correction. The encoder shares weights with Stage-I, while the decoder is separately optimized. The Stage-II decoder is initialized with the trained weights of the Stage-I decoder. Training is performed with a batch size of 4 and a learning rate of 1e-4 for 80k steps on eight NVIDIA H200 GPUs (approximately 1.5 days).

After training, they are integrated into a unified inference pipeline. For a given input image, it is first processed by the Stage-I Refocus Inpainter to perform inpainting. The generated content is then refined by the Stage-II Seam- and Shadow-Aware Decoder and blended back into the original image, yielding a final output that is visually coherent and free of seams or shadow inconsistencies.

## E    ADDITIONAL VISUALIZATIONS

To provide further qualitative insights into the effectiveness of ReFocusEraser, we present supplementary visual comparisons and ablation studies in this appendix.

**Visual Comparison Against Other SOTA Methods.** Figures 12 and 13 provide additional visual comparisons with state-of-the-art specialized object removal and general inpainting methods across diverse scenes. Our approach consistently produces semantically faithful and visually coherent results, excelling at recovering fine details of small objects and eliminating residual shadows, even in challenging scenarios with complex textures.

**Visual Impact of the Refocus Inpainter Trained at Different Focal Lengths.** Figure 14 provides additional visualizations demonstrating the effect of training the LoRA module at different focal lengths ($1\times$, $2\times$, $3\times$). Models trained at higher focal lengths capture finer structural details and context information, producing more accurate and visually coherent inpainting. These results offer further evidence of the benefits of focal-length-specific training.

**Visual comparison of fine-tuning the Flux.1-dev decoder with different losses.** Figure 15 provides additional visualizations that highlight the effectiveness of our proposed color-consistency loss. By applying L1 to foreground and LPIPS to background, our proposed loss not only preserves fine foreground details and mitigates color shifts to improve seamless integration, but also automatically removes residual shadows, producing more natural and visually coherent results.

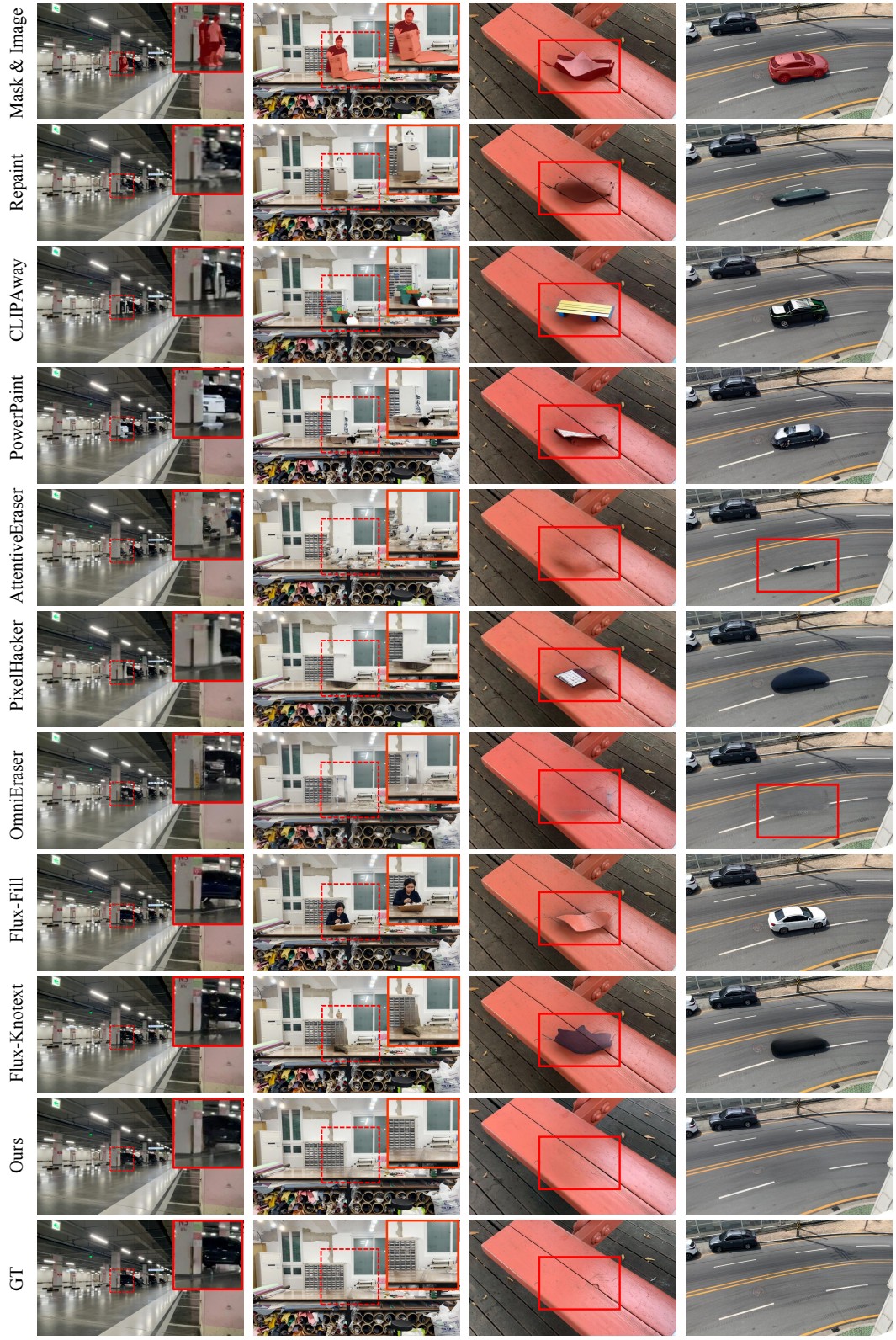

Figure 12: **Additional qualitative comparisons with state-of-the-art methods.** The red solid box shows a zoom-in view of the corresponding red dashed box. All visual results demonstrate the superiority of our method over existing approaches.

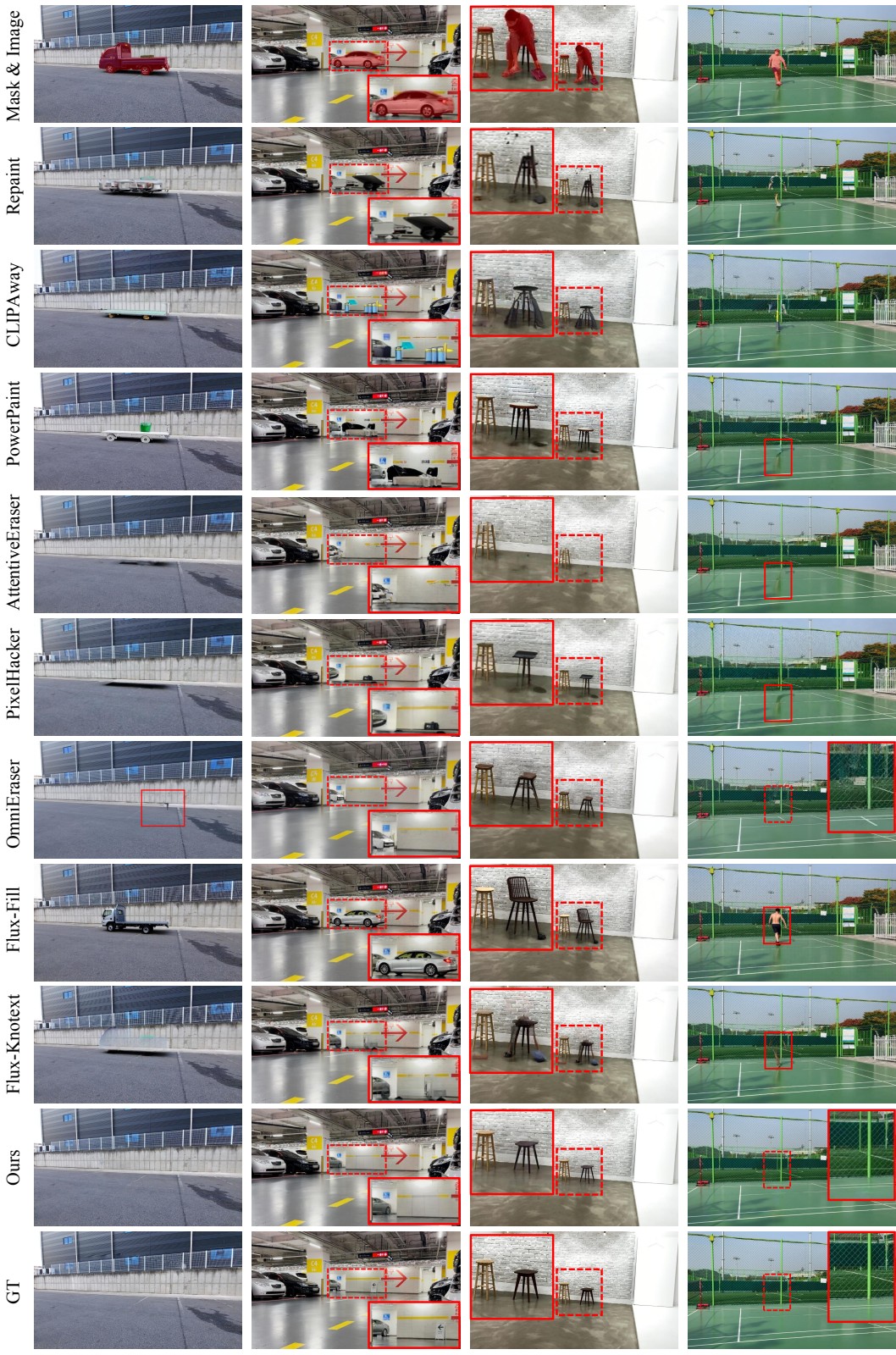

Figure 13: **Additional qualitative comparisons with state-of-the-art methods.** The red solid box shows a zoom-in view of the corresponding red dashed box. All visual results demonstrate the superiority of our method over existing approaches.

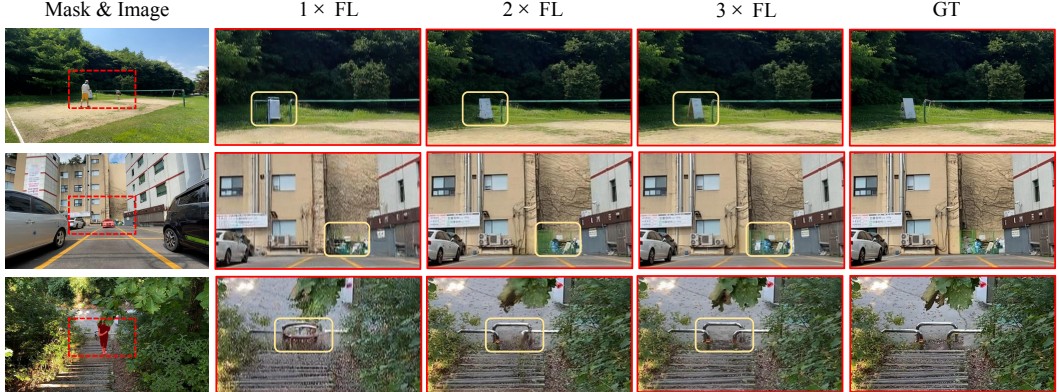

Figure 14: **Additional qualitative comparisons of the camera-adaptive refocus inpainter at different focal lengths** ($1\times$, $2\times$, **and** $3\times$)**.** Increasing zoom levels improve inpainting performance for large field-of-view images, with the $3\times$ producing the most accurate and visually coherent results.

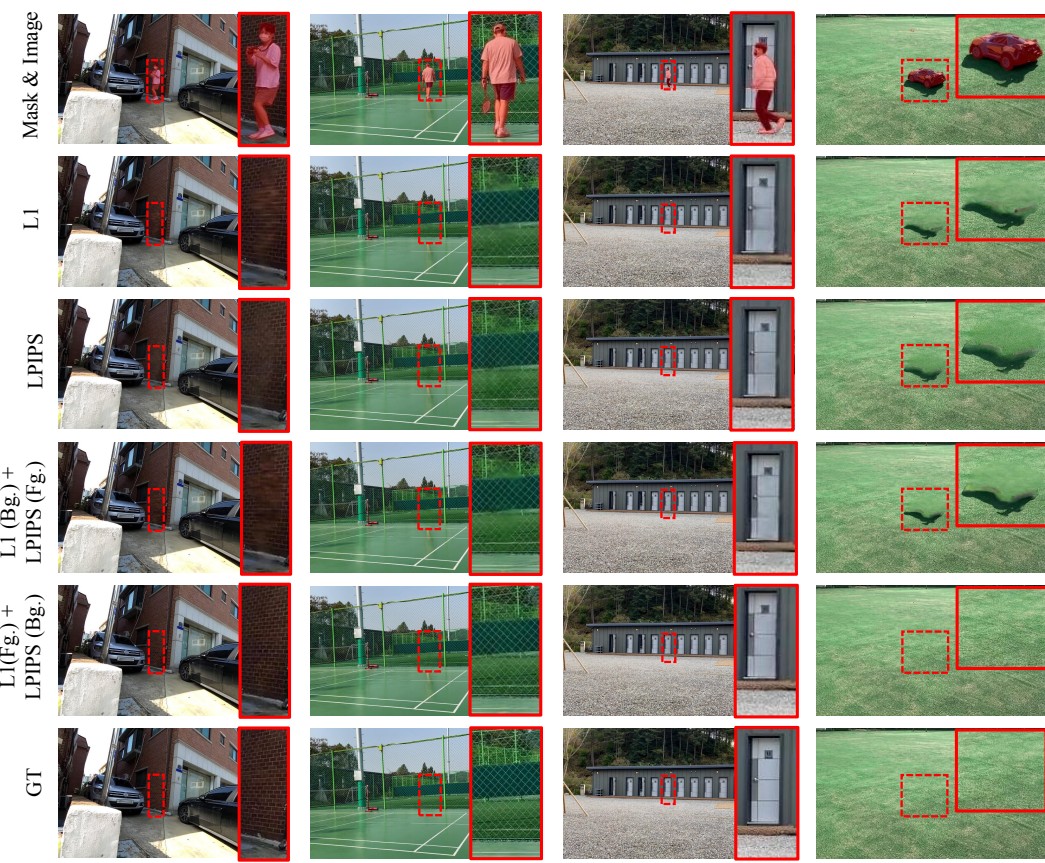

Figure 15: **Visual comparison of fine-tuning the Flux.1-dev decoder with different losses.** Using LPIPS for the background and L1 for the foreground not only ensures visual consistency to address seam issues caused by color shifts but also effectively repairs object shadows.

(a) Mask&Image     (b) OmniEraser     (c) Ours     (d) GT

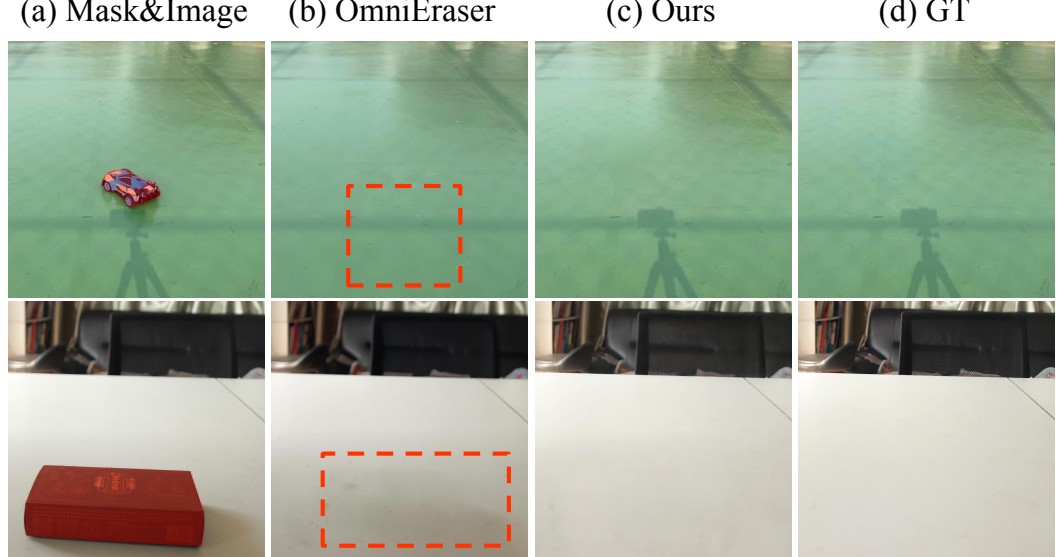

Figure 16: Comparison between OmniEraser and our method on shadow–object removal. While both methods remove the masked shadow, our results better preserve unrelated shadows in the scene and produce more complete shadow removal in the target region.

Table 6: Comparison of inference time, parameter count, and computational cost (GFLOPs) across different inpainting methods.

| Method | Inference Time(s) | # Parameters(B) | GFLOPs |
|---|---|---|---|
| Repaint | 173.58 | 0.55 | 554K |
| PixelHacker | 1.94 | 0.95 | 30K |
| CLIPAway | 3.65 | 1.09 | 84K |
| PowerPaint | 7.76 | 1.07 | 198K |
| Omnieraser | 9.07 | 16.96 | 1709K |
| Flux-Kontext | 33.02 | 16.87 | 3152K |
| Flux-Fill | 30.79 | 16.87 | 3066K |
| AttentiveEraser | 15.60 | 3.47 | 562K |
| ReFocusEraser | 12.28 | 17.01 | 1723K |

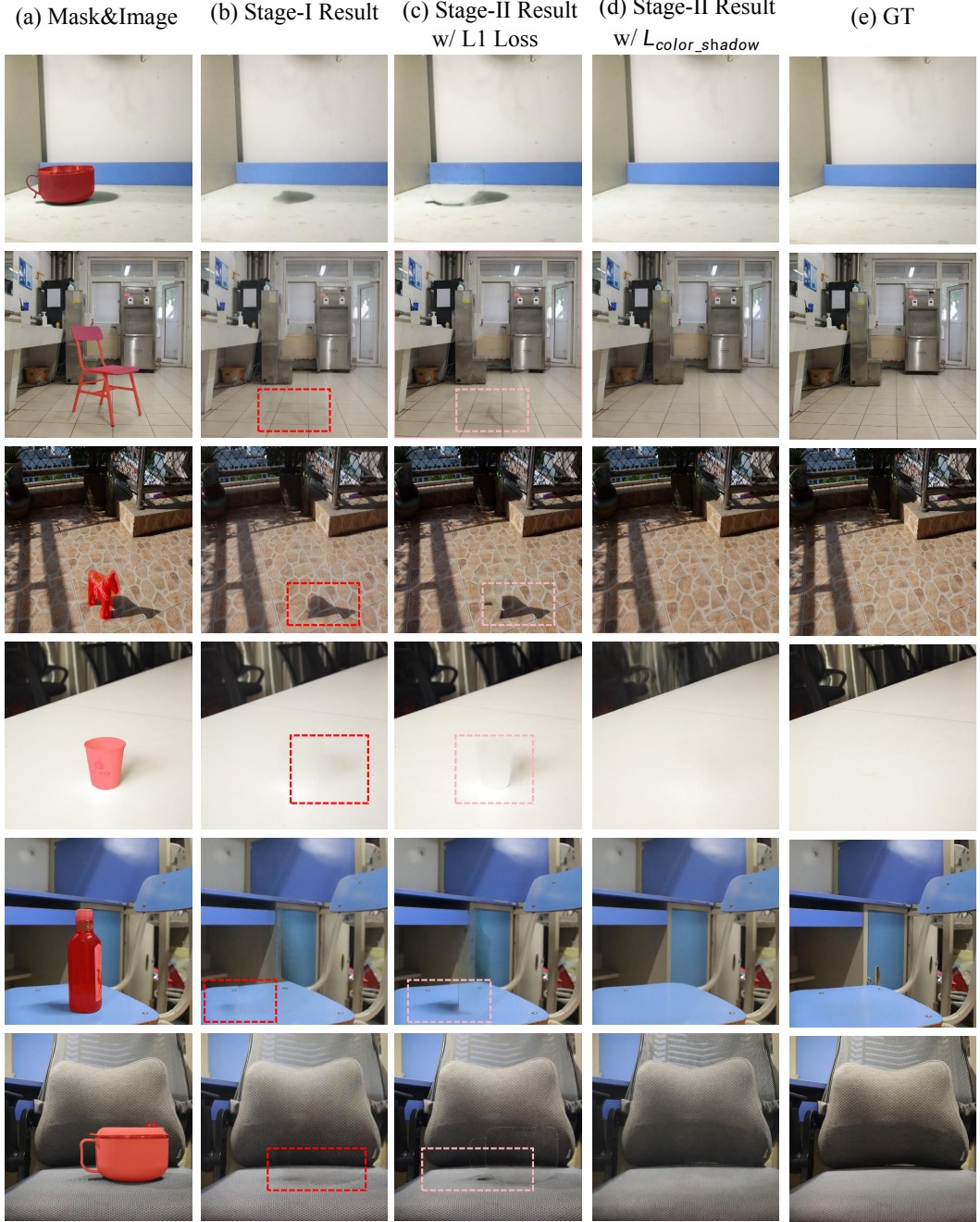

Figure 17: Comparison of shadow removal results. From left to right: input with mask, Stage-I output, Stage-II output, L1-VAE decode, OmniEraser, and ground truth. The two-stage model progressively improves structural recovery and texture consistency.

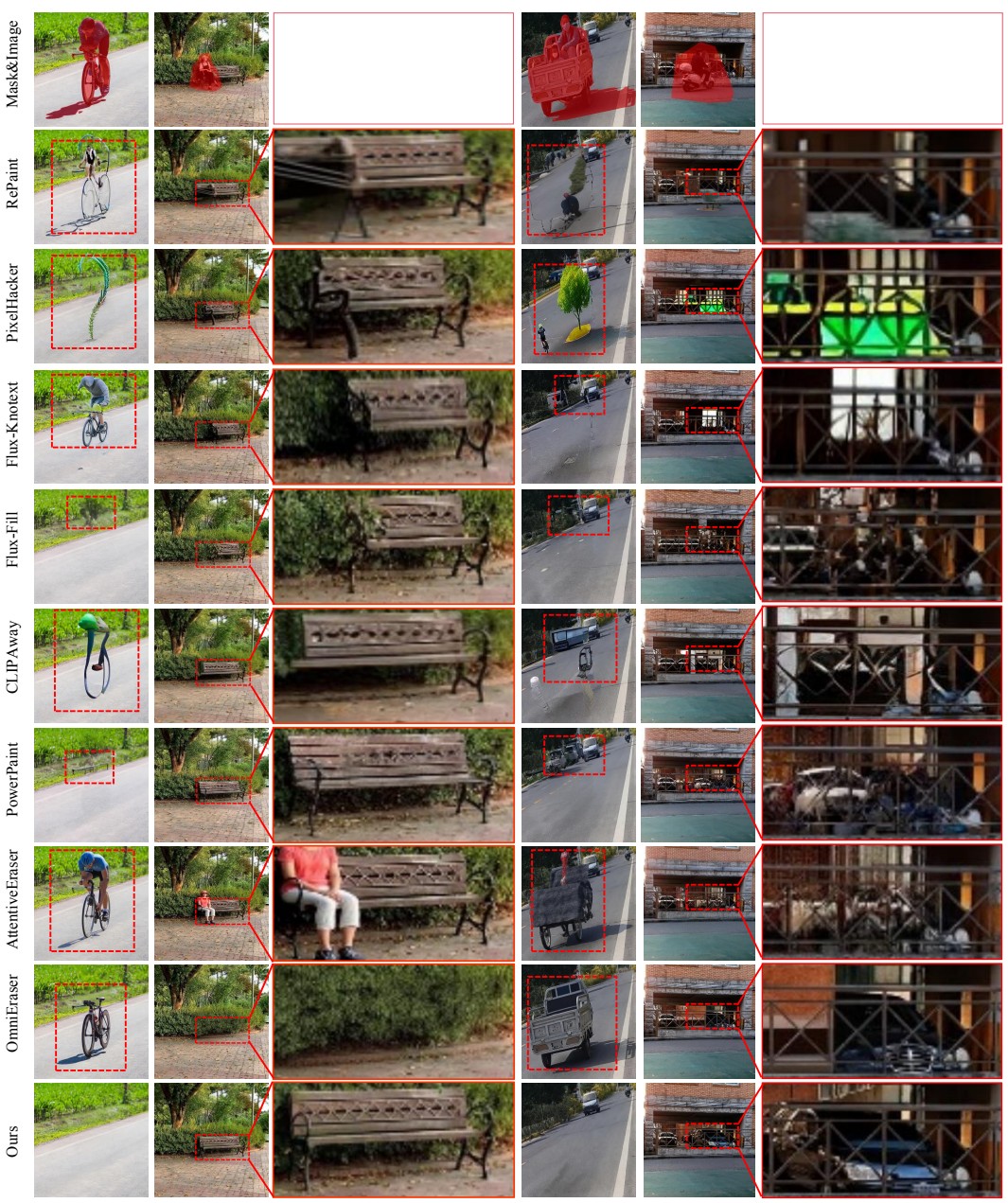

Figure 18: Fair comparison with object and shadow masks provided to all methods.

| Input Image | Reconstructed | Stage-I Result | Reconstructed |
| --- | --- | --- | --- |

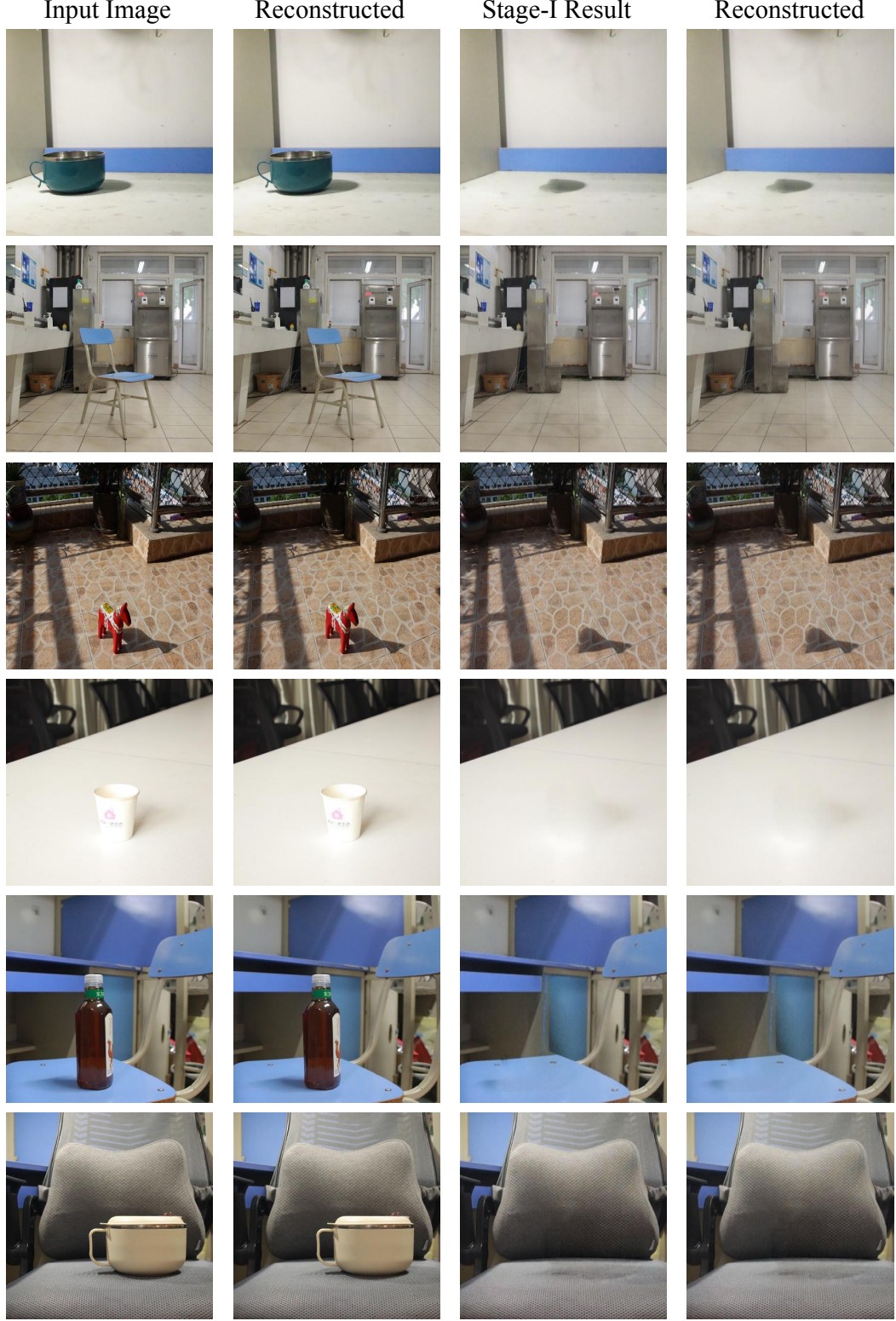

Figure 19: Demonstration of the strong reconstruction capability of the VAE.

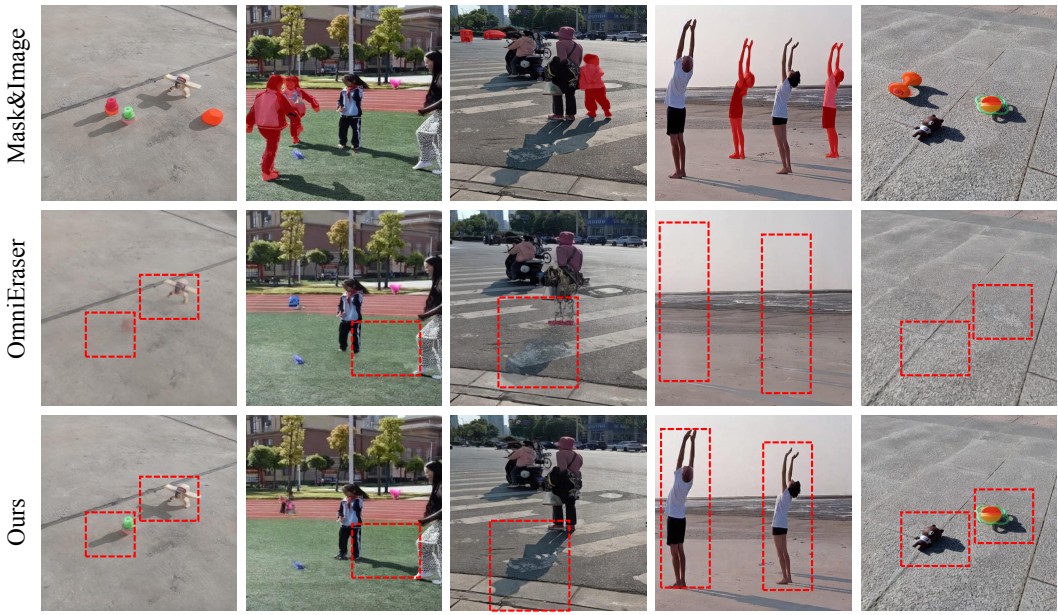

Figure 20: Our method removes only the target shadows while preserving shadows cast by other objects, demonstrating its robust selectivity.

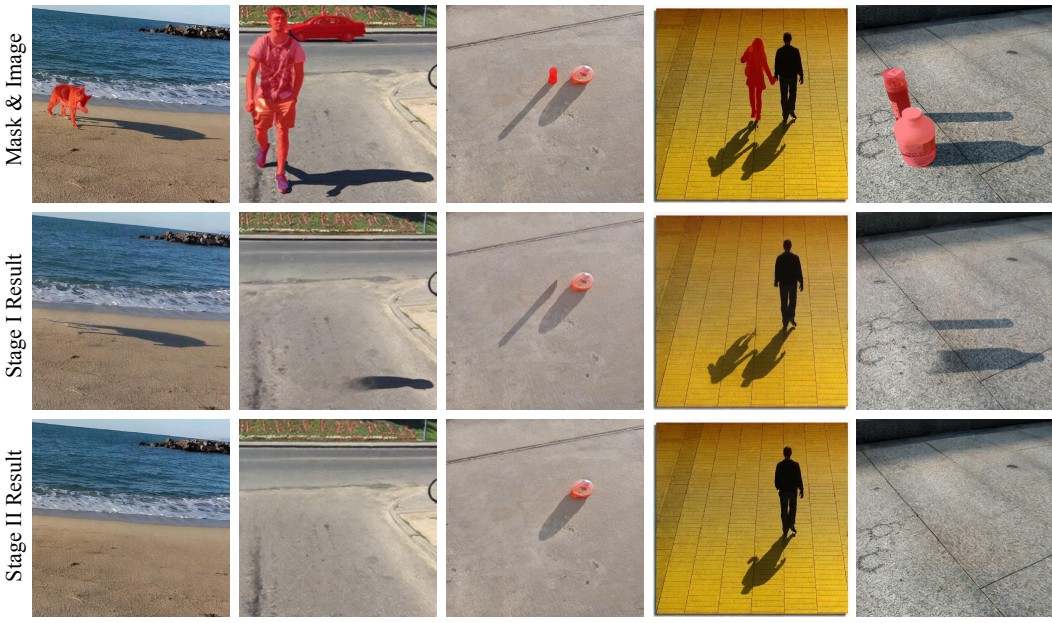

Figure 21: Comparison of Stage-I and Stage-II results on cases where the shadow region is spatially distant from the corresponding object. Each column shows the masked input, the intermediate Stage-I output, and the final Stage-II refinement.

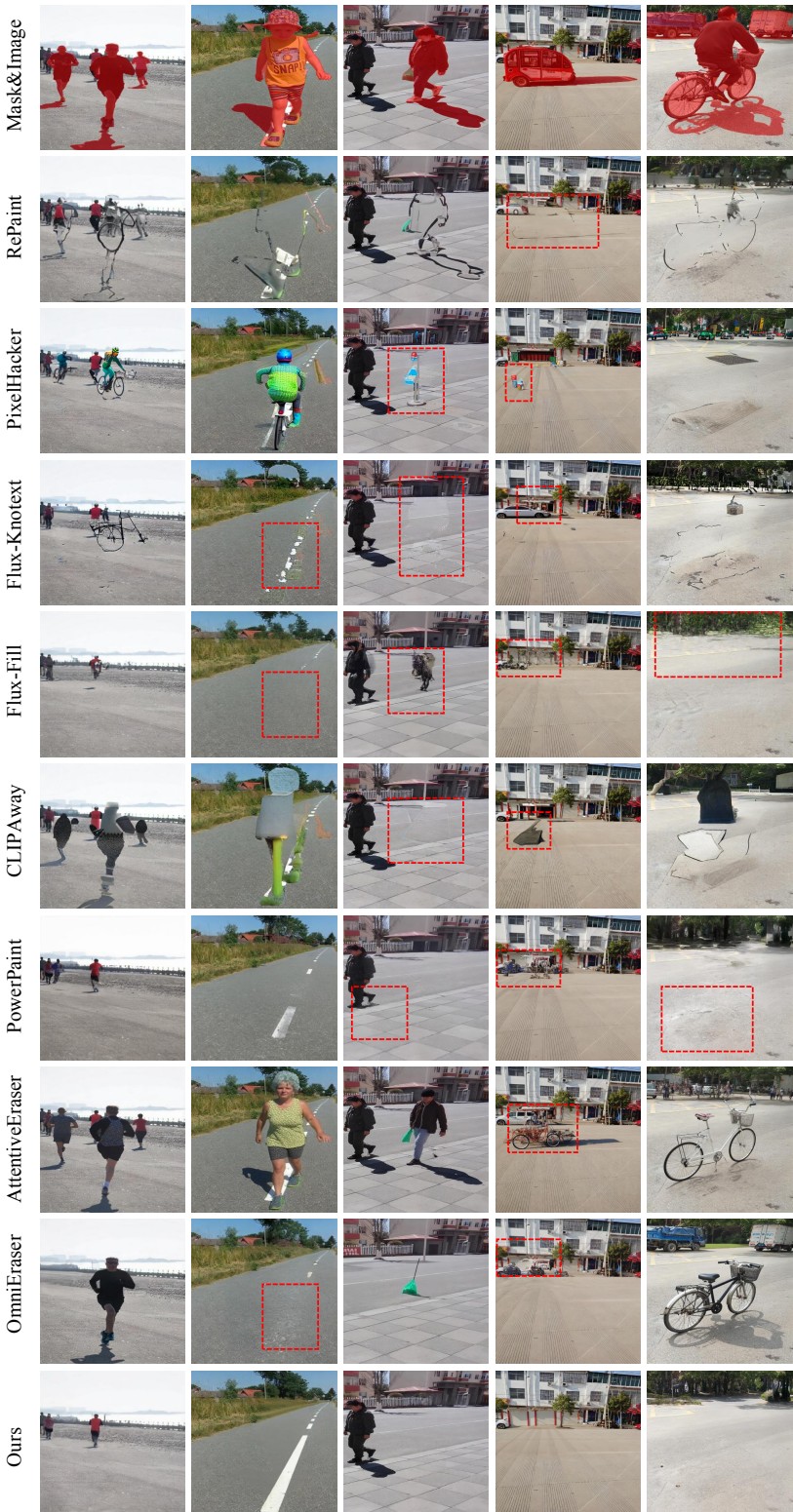

Figure 22: Compared with existing inpainting models, our method achieves stable and coherent removal of large objects, avoiding the severe distortions and missing textures observed in competing approaches.

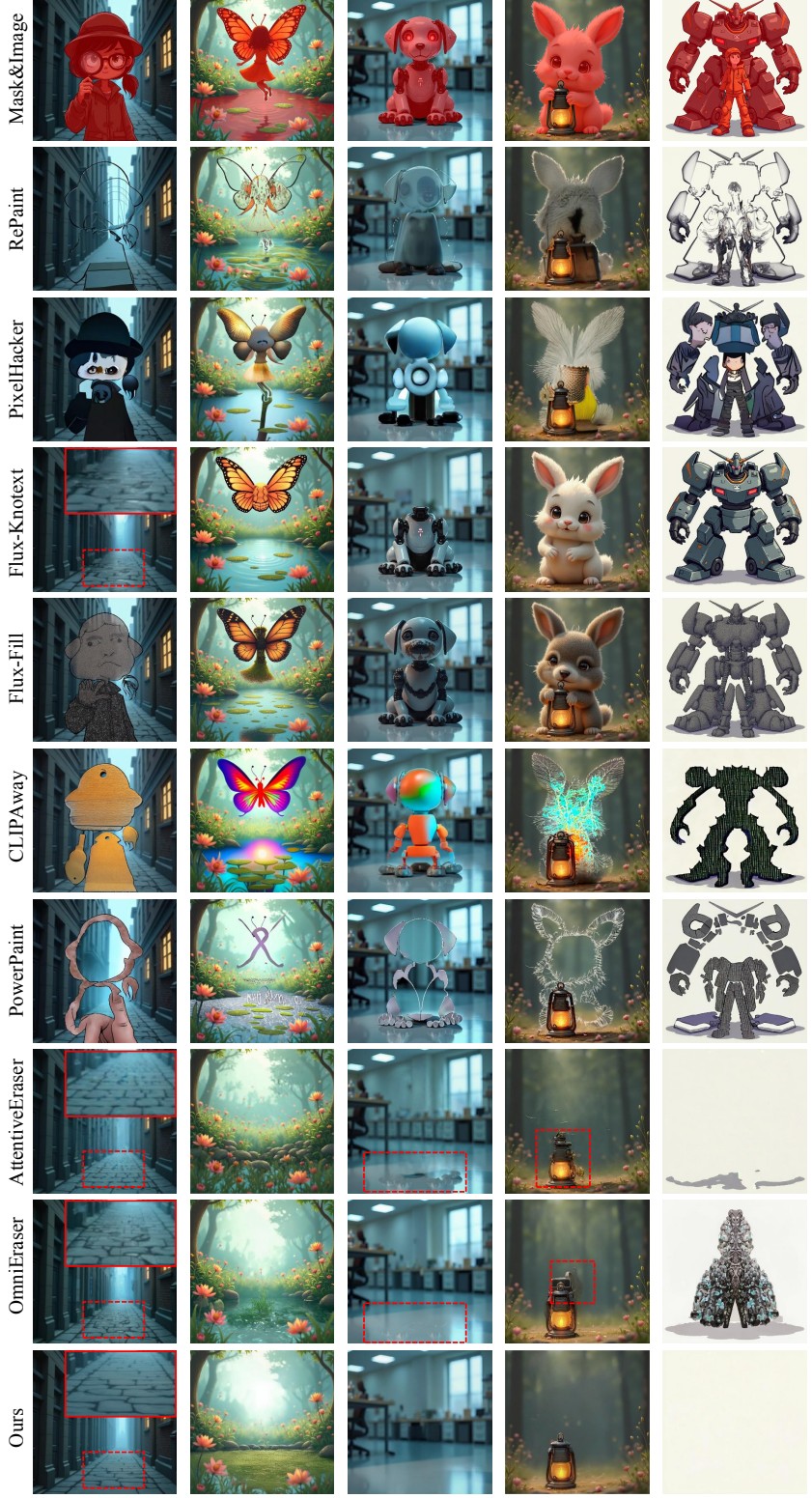

Figure 23: Examples of testing on non-camera images.

