# OpenReview forum: "ReFocusEraser: Refocusing for Small Object Removal with Robust Context-Shadow Repair"
_ICLR.cc/2026/Conference — ICLR 2026 Poster_

### Official Review · Reviewer_BXbz · 2025-10-17

**Soundness:** 2
**Presentation:** 3
**Contribution:** 3
**Rating:** 6
**Confidence:** 3

**Summary:**

This paper proposes ReFocusEraser, an innovative two-stage framework for small masked object removal with robust context and shadow repair.

**Strengths:**

1. This paper proposes an innovative two-stage framework for small masked object removal with robust context and shadow repair. By integrating a Camera-Adaptive Refocus Inpainter with a Seam- and Shadow-Aware Repair module, the method effectively enhances the structural and textural restoration of small masked regions.
2. Experimental results demonstrate that the proposed method achieves significant performance improvements on benchmark datasets such as RORD-val and RemovalBench, outperforming existing state-of-the-art methods.

**Weaknesses:**

1. Regarding the generalization of ReFocusEraser, this method is mainly evaluated on camera-captured data and scenes. It is unclear whether it remains effective on non-camera-captured images.
2. The current framework adopts a two-stage decoupled training strategy. If the two stages are trained jointly in an end-to-end manner, what are the differences in training cost and performance?
3. While training details are reported, the paper does not provide a discussion or comparison of inference efficiency. The overall training time, inference time, and computational complexity compared to mainstream methods remain unclear.

**Questions:**

Refer to "Weaknesses".

---

> ### Author Response · Authors · 2025-11-24
> **ReFocusEraser: Refocusing for Small Object Removal with Robust Context-Shadow Repair**
>
> Thank you to the reviewer for recognizing the innovation of our two-stage framework, which effectively addresses small object removal with robust context and shadow repair. We also appreciate the acknowledgment of our strong experimental results showing significant improvements over state-of-the-art methods on benchmark datasets. Below are our responses to the reviewer’s questions.
>
> > Question 1: Whether it remains effective on non-camera-captured images.
>
> **Response:**
> We generated prompts with ChatGPT and used them to create synthetic images via the Flux T2I model. Figure 23 compares our method with others on this synthetic dataset, demonstrating superior removal of regions and objects.
>
> > Question 2: What if the two stages are trained jointly in an end-to-end manner?
>
> **Response:**
> Our Stage-I outputs must be processed by the mask-stitching operation before being fed into Stage-II, which makes separate training of the two stages more practical. Joint end-to-end training is not feasible because it would require incorporating the non-differentiable mask-stitching step into the training pipeline. As illustrated in Figure 2, the Stage-I decoder is fixed, and no gradient propagation occurs between the decoders of Stage-I and Stage-II. Therefore, joint optimization of the two stages cannot be performed.
>
> > Question 3: Compare the inference time and computational complexity with other methods?
>
> **Response:**
> Table 6 compares inference time, parameter count, and GFLOPs of our method and other approaches. Methods like PixelHacker and CLIPAway offer faster inference and fewer parameters but generally underperform compared to Flux-based models. Flux-based approaches, including Flux-Kontext, Flux-Fill, and OmniEraser, deliver the strongest overall performance but require model sizes of around 17.01B parameters. Compared to existing Flux-based methods, our approach improves performance while adding only a small number of parameters (ours 17.01B vs. OmniEraser's 16.96B  ).

---

> > ### Comment · Reviewer_BXbz · 2025-11-26
> >
> > Thank you for providing the additional experiments and visualization results. They have addressed most of my questions and concerns.
> > However, if end-to-end joint training removes the mask-stitching process from the pipeline, doesn’t this reduce the complexity of the multi-stage processing? Although this might introduce some additional computational cost, it still seems to simplify the pipeline.

---

> ### Author Response · Authors · 2025-11-28
> **why it can not end-to-end joint training and the necessary of mask-stitiching**
>
> Thank you for your question. We made significant efforts to implement joint training but found it to be ineffective. Below, we provide a more detailed explanation.
>
> As illustrated in Figure 2, the decoder in Stage-I remains fixed, and no gradient propagation occurs between the decoders of Stage-I and Stage-II. While Stage-I aims to predict noise added during the forward diffusion process, Stage-II focuses on enhancing denoised images by correcting color discrepancies and shadow artifacts. The fundamentally distinct objectives of the two stages make joint end-to-end training infeasible.
>
> In the first stage, we inject LoRA into the Flux transformer while keeping the VAE of Flux.1-dev frozen. LoRA fine-tuning in Stage-I follows the diffusion training paradigm, where the model learns to predict noise added during the forward diffusion process rather than directly reconstructing clean data. In other words, Stage-I focuses on learning the noise-injection process.
>
> In contrast, Stage-II depends on the reverse diffusion process of Stage-I, which reconstructs clean content from noisy inputs, processes it using mask-stitching, and provides the resulting output as input to Stage-II. Regardless of whether mask-stitching is applied, the reverse diffusion output from Stage-I is always required. Stage-II then fine-tunes the decoder of Flux.1-dev to correct color discrepancies and mitigate shadow artifacts.
>
> The specific training time and GPU memory usage are shown in the table 1 below.
> ### Table 1: Training Time and GPU Memory Usage of Different Stages
>
> |       Module        |       Training Time       | Batch Size | GPU Memory Usage (GB) |  number of NVIDIA H200 GPUs|
> |:-------------------:|:-----------------------------------------------:|:----------:|:--------------------:| :--------------------:|
> |      Stage-I       |                 8 hrs 44 mins                    |     2      |        100.72        | 4|
> |      Stage-II       |                35 hrs 29 mins                    |     4      |        121.97        | 8|
>
> Furthermore, to assess the impact of using or not using the mask-stitching operation, we compare two Stage-II VAE variants: (1) the Stage-I output is directly used as the Stage-II input without mask-stitching, and (2) the Stage-I output is first processed by the mask-stitching operation before being fed into Stage-II. Apart from this difference in input, the training configurations of the two VAEs are kept strictly identical. We train two corresponding models under these settings to further verify the effect of mask-stitching strategy. The experimental results are reported in the table 2 below.
>
> ### Table 2: Performance comparison of Stage-II VAE variants with vs. without mask-stitching
> |      Method       |  PSNR $\uparrow$  | SSIM $\uparrow$  | LPIPS $\downarrow$  |  FID $\downarrow$  | CMMD $\downarrow$  |
> |:-----------------:|:------:|:-----:|:-----:|:------:|:-----:|
> | Stage-II w/o mask-stitching | 30.825 | 0.918 | 0.063 | 23.569 | **0.262** |
> | Stage-II w/ mask-stitching (Ours Stage-II)        | **31.256** | **0.924** | **0.041** | **21.379** | 0.263 |
>
> As shown in the table, the model trained with mask-stitched inputs achieves noticeably better reconstruction performance than the model trained without mask-stitching, further demonstrating the effectiveness of the proposed mask-stitching strategy.

---

### Official Review · Reviewer_5SDG · 2025-10-25

**Soundness:** 3
**Presentation:** 3
**Contribution:** 2
**Rating:** 6
**Confidence:** 4

**Summary:**

This paper proposes diffusion-based framework to handle *object removal*, capable of handling objects of various sizes. To improve the object removal of small objects with securing the consistency between inpainted region and surrounding context, the paper proposes two main components:

1. Camera-adaptive zoom-and-crop module to upscale the appearance of small objects.
2. Additional decoder to mitigate the misalignment between surrounding context and removed region.

**Strengths:**

* The paper points out the limitation of previous works that struggle to remove the small objects in the image due to the information loss in VAE encoder. The paper addresses aforementioned issue where the motivation is clear.
* The paper proposes *Camera-adaptive Refocus Mechanism* where it crops the target object based on the camera parameters, better preserving the object context. Furthermore, it introduces additional decoder to enhance the inconsistency between removed part and surrounding context such as boundary of the removed object. The framework makes sense with resulting in clear performance improvement compared to the baseline.

**Weaknesses:**

My main concern is on the efficiency of the model.

* **Training Efficiency:** In L47-49, the paper points out that fine-tuning the VAE incurs huge computational cost.  However, ReFocusEraser also needs substantial amounts of GPUs which is 8 H200 GPUs to optimize the additional decoder.
* **Inference/parameter Efficiency:** ReFocusEraser needs two decoder for the inference which raises the concern on the efficiency of inference time and parameter size. Can you compare the inference time of ReFocusEraser with the baselines in Tab. 1? Also, it would be also better to compare the size of parameters as well.
* In Fig. 6b and Fib. 6d, it seems that there are color shift (more brighter) on the removed part compared to the Fig. 6a and Fib. 6c. Due to this color shift, it seems that box-based stitching results in more natural output compared to the mask-based stitching which is inconsistent with the paper's conclusion on L431.
* In Tab. 2, PSNR of row (c) is higher than row (d). However, the number on row (d) is bolded. Furthermore, not using decoder achieves better results on PSNR, SSIM and LPIPS compared to the setting with using additional decoder while opposed results can be found in FID and CMMD metrics. Can you explain more about this inconsistent results?
* In Tab. 3, it is better to add the results with fixed crop or zoom to validate the effectiveness of camera-adaptive zoom-in method.

**Questions:**

NA

---

> ### Author Response · Authors · 2025-11-24
> **ReFocusEraser: Refocusing for Small Object Removal with Robust Context-Shadow Repair**
>
> Thank you to the reviewer for recognizing that our paper clearly identifies the challenges past methods face in removing small objects due to information loss in the VAE encoder, and for acknowledging how our approach addresses this. We also appreciate the valuable suggestions. Below are our responses.
>
> > Question 1: Explain training efficiency for the additional decoder.
>
> **Response:**
> Finetuning the entire VAE can lead to severe catastrophic forgetting of the pretrained model’s capabilities, and addressing this issue would require substantially more training data, longer training time, and significantly higher computational cost [2]. To avoid this problem, we fine-tune only the decoder, and our mask-stitching strategy further limits model updates to the masked regions, reducing the risk of forgetting.
>
> The additional decoder is lightweight (only 0.05B parameters; see Table 6, final model 17.01B vs. baseline 16.96B) and can be trained on a single GPU (we used eight H200 GPUs only to accelerate training). This design also keeps the encoder identical in both Stage-I and Stage-II, so inference only requires loading the additional decoder parameters.
>
> Details of the modifications have been incorporated into the revised manuscript (lines 47–49 and lines 215–221).
>
> [2] Otani A. Mitigating Catastrophic Forgetting and Mode Collapse in Text-to-Image Diffusion via Latent Replay[J]. arXiv preprint arXiv:2509.10529, 2025.
>
>
> > Question 2: Compare inference/parameter efficiency in Table 1.
>
> **Response:**
> Table 6 compares the inference and parameter efficiency of our method and other approaches. Our additional decoder only adds 0.05B, compared to other Flux-based approaches, but deliver the strongest overall performance.
>
> > Question 3: Box-based stitching appears more natural than mask-based stitching, contradicting the paper’s conclusion on L431.
>
> **Response:**
> At line 431, we claim that mask-stitching preserves the background appearance better than box-stitching. Figure 6 in the original manuscript presents Stage-I outputs with different stitching strategies applied for comparison. As shown in (a) and (c), box-stitching introduces noticeable color inconsistencies between the stitched region and the background, whereas in (b) and (d), such inconsistencies are confined to the masked area. Therefore, the conclusion that mask-stitching better maintains background fidelity remains valid.
>
> > Question 4: Can you explain more about the inconsistent results in Table 2 (c) and (d)?
>
> **Response:**
> The bolded region was mislabeled, and we have corrected it in the revision. In Stage-II, the decoder corrects color and shadow artifacts, improving perceptual realism. However, because the VAE in Flux.1-dev is not perfectly lossless, it introduces slight pixel-level deviations, leading to lower PSNR, SSIM, and LPIPS, while FID and CMMD still improve in row (d).
>
> > Question 5: In Tab. 3, it is better to add the results with fixed crop or zoom to validate the effectiveness of camera-adaptive zoom-in method.
>
> **Response:**
> In Table 3, “1X” represents the method without cropping, while “2X” and “3X” correspond to cropping based on the camera’s focal length. This approach validates the effectiveness of the camera-adaptive zoom-in. Fixed cropping methods cannot reliably distinguish between small and large targets since they require manual assessment for each image, making them impractical. In contrast, using focal length to determine target size offers a more systematic and scalable approach.

---

### Official Review · Reviewer_Prd8 · 2025-10-30

**Soundness:** 2
**Presentation:** 3
**Contribution:** 2
**Rating:** 6
**Confidence:** 2

**Summary:**

This paper proposes a two-stage framework for small object removal, addressing the performance limitations caused by detail loss in the VAE encoder’s downsampling process. In the first stage, the authors introduce a camera-adaptive refocus mechanism that enlarges masked regions to facilitate coarse object removal. In the second stage, they fine-tune an additional decoder to correct residual artifacts such as color shifts and seams. The method is evaluated through experiments on both small and large object removal tasks.

**Strengths:**

1. The paper effectively addresses the challenging and often overlooked problem of small object removal.
2. The work is supported by extensive experiments and demonstrates compelling qualitative results.
3. The paper is well-structured and clearly presented.

**Weaknesses:**

1. The camera-adaptive refocus mechanism in Stage-I appears highly effective for small objects, but its contribution to large object removal is less clear. An ablation study specifically isolating the contributions of Stage-I and Stage-II for large object removal would be insightful.
2. For Stage-II, it is unclear how robust the shadow-aware repair is to shadows that are cast far from the original mask region. Could such cases lead to failures or the introduction of new artifacts?
3. In Table 2, the metrics for Exp. (d) show a slight degradation in PSNR, SSIM, and LPIPS compared to Exp. (c). Could the authors provide further analysis on the reason for this performance trade-off?
4. Including a discussion or visualization of typical failure cases would further strengthen the paper by clarifying the method's limitations.

**Questions:**

Please refer to the weaknesses.

---

> ### Author Response · Authors · 2025-11-24
> **ReFocusEraser: Refocusing for Small Object Removal with Robust Context-Shadow Repair**
>
> Thank you the reviewer #Prd8 for recognizing our paper’s effective handling of small object removal and the strong experimental support. We appreciate your positive feedback on the organization and clarity of the paper, as well as your detailed comments, which we address below.
>
> > Weakness 1: Contribution to large object removal.
>
> **Response:**
> In the first stage, only the LoRA is trained while the Flux.1-Dev VAE remains fixed. This LoRA effectively handles the removal of both small and large target objects. During inference, we adaptively determine whether the target object is small based on its focal length, ensuring that the removal process for large objects remains unaffected.
>
> Figure 22 demonstrates that our full pipeline outperforms existing methods in removing large target objects, confirming its effectiveness across both small and large object scenarios.
>
>
> > Weakness 2: How robust the shadow-aware repair is to shadows that are cast far from the original mask region?
>
> **Response:**
> Figure 21 visually compares shadow restoration results between Stage-I and Stage-II, showing that Stage-II achieves more effective removal of the target’s shadow while preserving shadows of non-target objects. Notably, the results in the second row of Figure 21 correspond to the zoomed-in regions whose Stage-I outputs are used directly without any mask-stitching processing. As shown by the visualizations in the second row of Figure 2, the Stage-I model exhibits limited shadow-removal capability.
>
> In contrast, Stage-II uses inputs processed with mask-stitching and incorporates the shadow removal loss introduced in our paper, resulting in significantly improved shadow elimination.
>
> Moreover, as addressed in our response to Reviewer #bExM’s Question 2, although the VAE-based process is not entirely lossless, the resulting changes are visually negligible to the human eye, as illustrated in Figure 19.
>
> > Weakness 3: Table 2 shows Exp. (d) has slightly lower PSNR, SSIM, and LPIPS than Exp. (c). Explain why?
>
> **Response:**
> The slight decrease in PSNR, SSIM, and LPIPS observed in Experiment (d) compared to (c) is caused by the presence of shadows on objects in the test dataset. These shadows introduce noticeable color inconsistencies near the mask edges when the mask-stitching strategy is applied to paste the edited region back into the original image.
>
> In Stage-II, our decoder is specifically finetuned to correct such color discrepancies and shadow artifacts. While the Flux.1-dev VAE demonstrates strong reconstruction capabilities with only minor pixel-level differences that are visually negligible, the overall reconstruction process is not perfectly lossless.
>
> Consequently, although experiment (d) shows a slight drop in pixel-level metrics, it achieves a significantly better FID score, which more accurately reflects the overall visual quality (as shown in Figure 4(c) and (d)).
>
>
> > Weakness 4: Discussion limitation.
>
> **Response:**
> One limitation of our method is that, although the decoder in the second stage is trained to reduce color differences and shadow artifacts, the VAE in Flux.1-dev is not perfectly lossless.

---

### Official Review · Reviewer_bExM · 2025-10-31

**Soundness:** 2
**Presentation:** 3
**Contribution:** 2
**Rating:** 4
**Confidence:** 4

**Summary:**

This paper proposes a two-stage framework to address the small object removal problem. First, the small object is enlarged using the Camera-Adaptive Refocus Strategy and input into the first-stage Camera-Adaptive Refocus Inpainter, where the object is removed via a fine-tuned pre-trained FLUX model. After that, the second-stage Seam- and Shadow-Aware Repair is applied. This step uses a fine-tuned FLUX VAE to maintain background consistency and eliminate shadows. Both metric validation and visual result demonstrations confirm the effectiveness of this framework.

**Strengths:**

1. This method can address the small object removal challenge and the color shift issue inherent in diffusion models.
2. The framework not only achieves significant metric improvement over most SOTA models but also produces high-quality visual results.

**Weaknesses:**

1. The first stage is reported to remove shadows, yet the shadowy background is pasted back and shadow removal is repeated in the second stage. This process seems to waste the shadow-free result obtained from the first stage.
2. In the experimental comparison with other models that lack the ability to remove unmasked shadows, providing only object-only masks may introduce unfairness to these competing methods.

**Questions:**

1. It is noted that RORD’s masks include objects, shadows, and an extra expanded edge, whereas the masks presented in this work only fit tightly around objects. It remains unclear whether these masks were re-labeled, and whether the same "object-only" masks were used for model training.
2. The reason for selecting VAE for the second-stage correction requires further clarification, as other architectures may exhibit better performance. Additionally, VAE typically tends to reduce image quality, and how this issue is addressed in the proposed method needs more explanation.
3. In the second stage, since only the zoomed image (with shadows but without the target object) is input, the model may struggle to distinguish which shadow originates from the target object. It is recommended to provide test results on a more challenging scenario: when other non-target-related shadows exist in the zoomed image, can the method exclusively remove the target’s shadow without affecting the others?
4. Whether the Camera-Adaptive Refocus Strategy has considered the possibility that part of the object’s shadow may be cropped outside the zoomed-in region needs to be confirmed.

---

> ### Author Response · Authors · 2025-11-24
> **ReFocusEraser: Refocusing for Small Object Removal with Robust Context-Shadow Repair**
>
> We thank the reviewer #bExM for their positive feedback and recognition of our method’s effectiveness in addressing small object removal and color shift, as well as its strong performance in both metrics and visual quality. We appreciate the detailed comments and address the points below.
>
> > Weakness 1: Waste the shadow-free result obtained from the first stage.
>
> **Response:**
> Although the first stage has some shadow removal capability, its limitations require a second stage to further enhance the shadow removal effect.
>
> Our baseline model follows OmniEraser by taking the foreground object and background images as separate input guidance. This design inherently provides a certain capability for shadow removal, as empirically validated in the original OmniEraser work.
> However, this stage has limited capability in shadow removal, as highlighted by the red dashed boxes in Figure 16.
>
> The second-stage VAE is fine-tuned with a color-shadow consistency loss that separately reconstructs the foreground and background, enabling it to perform significantly stronger shadow correction (as shown in Figure 17).
>
> > Weakness 2: How do the results compare when both object and shadow masks are provided to all methods?
>
> **Response:**
> As shown in Figure 18, using an object mask with or without its shadow mask, our method consistently outperforms prior approaches and achieves superior background restoration even under irregular mask shapes.
>
> > Question 1: Whether these masks were re-labeled, and whether the same "object-only" masks were used for model training?
>
> **Response:**
> We did not manually re-annotate the masks; instead, we directly utilized the object mask information provided in RORD. RORD includes semantic segmentation results, which we treat as object masks. When multiple masks correspond to the same object, we merge them into a single, unified mask. These details have been added in the supplementary material of the manuscript (lines 787–790).
>
> >Question 2: Further clarification is needed on choosing VAE for the second stage and its impact on image quality.
>
> **Response:**
> The VAE in Flux.1-dev already demonstrates strong reconstruction ability by scaling up the training compute and utilizing 16 latent channels, which preserve both spatial resolution and original pixel information [1]. Our experiments, some of which are presented in Figure 19, further validate the reconstruction capability of the Flux.1-dev VAE.
>
> Due to this capability, we opt to fine-tune the VAE’s decoder in the second stage with separate foreground and background losses to mitigate color inconsistencies introduced during mask stitching. This strategy maximally preserves pixel-level semantic invariance while still enabling effective correction of background shadows and foreground color shifts. Figures 7 and 17 in the paper provide visual evidence supporting this outcome.
>
> [1] S. Batifol, A. Blattmann, F. Boesel, S. Consul, C. Diagne, T. Dockhorn, J. English, Z. English, P. Esser, S. Kulal, et al., "Flux.1 Kontext: Flow matching for in-context image generation and editing in latent space," arXiv preprint arXiv:2506, 2025.
>
>
> >Question 3: When other non-target-related shadows exist, can the method exclusively remove the target’s shadow without affecting the others?
>
> **Response:**
> As shown in Figure 20 and Figure 21 (rows 1 and 3), even when the target object’s shadow coexists with shadows from other non-target objects in the zoomed image, our method accurately removes only the target’s shadow while preserving the shadows of other non-targets' objects. Moreover, to minimize the impact on non-target objects, we employ a mask-stitching approach that enables effective removal of the target’s shadow without noticeably affecting surrounding areas. Although this process is not entirely lossless, the resulting changes are visually negligible to the human eye, as demonstrated in Figure 19.
>
> >Question 4: Does the Camera-Adaptive Refocus account for shadows cropped outside the zoomed region?
>
> **Response:**
> Small objects are typically captured using short focal lengths, which provide a wider field of view, causing these objects to occupy a smaller portion of the image. In our method, cropping is controlled based on focal length to ensure the cropped region remains tightly focused on the target area. We classify objects with a focal length below 500 pixels as small targets, and those above 500 pixels as large targets. During inference, we check whether the removal object’s mask has been scaled proportionally to guarantee that the object stays within the crop region. If this condition is not met, the camera-adaptive zooming operation is skipped, and the full image is fed directly into the network.

---

### Meta-Review · Area_Chair_kTe7 · 2026-01-04

**Summary:**

This paper addresses the challenging problem of small object removal in images, which is often hindered by information loss during the VAE encoding process in diffusion models. The authors propose ReFocus Eraser, a two-stage framework featuring a camera-adaptive refocus mechanism to enlarge masked regions for better detail preservation, followed by a seam- and shadow-aware repair stage. While reviewers initially expressed concerns regarding the efficiency of the multi-stage pipeline , the necessity of the second stage , and the fairness of mask comparisons , the subsequent rebuttals provided significant experimental evidence and technical clarification that largely satisfied the panel. The method demonstrates state-of-the-art performance on benchmarks like RORD-val and RemovalBench.

Hence, the AC recommends acceptance.

**Reviewer Concerns:**

Reviewer bExM has concerns about the redundancy of the shadow removal. The authors clarified that while Stage I has certain shadow removal capability, Stage II is essential for further ensuring background consistency.

Reviewer BXbz and 5SDG both express worry about the efficiency. The authors pointed out that in the appendix table 6, they list the number of parameters, running time, and flops. Considering the performance gains, such a computational burden is tolerable.

Reviewer BXbz asks whether the two-stage process can be merged into one. The authors point out that two objectives can not be jointly trained.

**Reviewer Scores:**

Reviewer bExM: 4.
Reviewer Prd8: 6
Reviewer 5sDG: 6
Reviewer BXbz: 6

The AC thinks that the 5sDG and Prd8 would highly likely keep the score, and BXbz explicitly replied that the rebuttal had addressed his concerns, but put one more question, thus the review would like to increase the score (at least leaning more to accept the paper). Most of the concerns from bExM are addressed.

---

### Decision · Program_Chairs · 2026-01-26

Accept (Poster)